# A distinct topology of BTN3A IgV and B30.2 domains controlled by juxtamembrane regions favors optimal human γδ T cell phosphoantigen sensing

Mohindar M. Karunakaran [1] ✉, Hariharan Subramanian [2,3], Yiming Jin[4], Fiyaz Mohammed[5], Brigitte Kimmel[6], Claudia Juraske[7,8,9,10], Lisa Starick[1], Anna Nöhren[1], Nora Länder[1], Carrie R. Willcox [5], Rohit Singh [4,11,12], Wolfgang W. Schamel [7,8,9,10], Viacheslav O. Nikolaev [2,3], Volker Kunzmann[6], Andrew J. Wiemer [4,11], Benjamin E. Willcox [5] & Thomas Herrmann [1] ✉

Butyrophilin (BTN)−3A and BTN2A1 molecules control the activation of human Vγ9Vδ2 T cells during T cell receptor (TCR)-mediated sensing of phosphoantigens (PAg) derived from microbes and tumors. However, the molecular rules governing PAg sensing remain largely unknown. Here, we establish three mechanistic principles of PAg-mediated γδ T cell activation. First, in humans, following PAg binding to the intracellular BTN3A1-B30.2 domain, Vγ9Vδ2 TCR triggering involves the extracellular V-domain of BTN3A2/BTN3A3. Moreover, the localization of both protein domains on different chains of the BTN3A homo-or heteromers is essential for efficient PAg-mediated activation. Second, the formation of BTN3A homo-or heteromers, which differ in intracellular trafficking and conformation, is controlled by molecular interactions between the juxtamembrane regions of the BTN3A chains. Finally, the ability of PAg not simply to bind BTN3A-B30.2, but to promote its subsequent interaction with the BTN2A1-B30.2 domain, is essential for T-cell activation. Defining these determinants of cooperation and the division of labor in BTN proteins improves our understanding of PAg sensing and elucidates a mode of action that may apply to other BTN family members.

Vγ9Vδ2 T cells comprise 1–5% of human peripheral blood T cells. They are massively expanded in some infections and exert multiple effector functions such as perforin-mediated cell lysis, help for other immune cells and peptide antigen-presentation. These functions are instrumental in the control of infection and tumors. Consequently, they have become the subject of an increasing number of preclinical and clinical studies[1–3].

Vγ9Vδ2 T cell receptors (TCRs) contain a semi-invariant γ chain with a Vγ9JP (alternatively termed Vγ2Jγ1.2) rearrangement and highly diverse Vδ2-bearing δ chains[4] and are activated by phosphoantigens (PAg) such as host-derived isopentenyl diphosphate (IPP) and microbially derived (E)−4-hydroxy-3-methyl-but-2-enyl diphosphate (HMBPP). In some tumors and infected cells, IPP levels reach a level sufficient to activate Vγ9Vδ2 T cells[5–8]. This activation can also be achieved pharmacologically by aminobisphosphonates (such as zoledronate), which inhibit the IPP-catabolizing farnesyl disphosphate synthase[5,9] or by farnesyl diphosphate synthase specific inhibitory RNA[10]. HMBPP is the immediate precursor of IPP in the non-mevalonate pathway of IPP

synthesis in many eubacteria, in apicomplexan parasites such as *Plasmodium* spp., and in chloroplasts. PAg activity of HMBPP is several orders of magnitude higher than that of IPP[11,12].

PAg-mediated activation of Vγ9Vδ2 T cells requires expression of butyrophilin 2A1 (BTN2A1)[13,14] and butyrophilin 3A1 (BTN3A1)[15] by target cells. Both molecules are single membrane-spanning type I proteins composed of a B7-like extracellular region comprising an N-terminal IgV-like (V) and a membrane-proximal IgC-like (C) domain, a transmembrane domain, and a cytoplasmic region comprising a juxtamembrane (JM) region and a B30.2 domain[16,17]. BTN2A1 binds with its V-domain to germline-encoded regions in the CDR2 and HV4 regions of the Vγ9-domain of the TCRγ chain[13,14] and to the V-domain of BTN3A1. The BTN3A1-B30.2 domain binds to PAg[18,19]. Furthermore, we and others showed that binding of the BTN3A1 B30.2 domain to PAg induces its binding to the B30.2 domain of BTN2A1[20,21], a process in which the JM regions of both molecules play a pivotal role. How these events finally translate into TCR-mediated Vγ9Vδ2 T cell activation is not yet understood[22] but evidence suggests that multiple CDRs of both the TCR-γ and -δ chains are involved, as evidenced by site-directed mutagenesis[23] and demonstration of interdependence of CDR3s from both chains in PAg reactivity[24].

*BTN3A* genes emerged with placental mammals but became defunct in many species, including mice and rats, similar to the co-evolving homologs of human Vγ9 (*TRGV9*) and Vδ2 (*TRDV2*) TCR genes[25]. The human *BTN3A* gene family comprises *BTN3A1, BTN3A2,* and *BTN3A3* and was generated by gene duplication events during primate evolution[26,27]. The gene products are expressed by most cell types, including αβ and γδ T cells. The PAg-binding site of BTN3A1 is a highly conserved, positively charged pocket formed by six amino acids of the intracellular B30.2 domain[18,28]. Upon PAg binding, this domain and the adjacent JM region undergo conformational changes[19,29–31] that are necessary for PAg-induced activation of Vγ9Vδ2 T cells.

Since their emergence in primates, BTN3A family members have diversified structurally and, most likely, functionally. Relative to BTN3A1, BTN3A2 lacks the entire B30.2 domain and parts of the JM region, while BTN3A3 bears an H381R substitution which abrogates PAg binding to the pocket (numbering of amino acids as in Supplementary Fig. 1a)[18]. The amino acid sequence identity of C-domains of the human BTN3As is about 90%, while the V domains of BTN3A1 and BTN3A2 are identical and that of BTN3A3 differs by a single conservative substitution (K66R) (Supplementary Fig. 1a)[22].

The contribution of BTN3A2 and BTN3A3 to PAg-mediated activation has been reported based on BTN3A family member knockdown studies in HeLa cells[32] and BTN3A knockout of 293T cells and various other cell lines[33–35]; consistent with this, we have observed superior PAg responses when BTN3A1 was re-expressed in BTN3A1KO (*BTN3A1* gene inactivated) cells than in BTN3KO cells in which all three *BTN3A* genes are inactivated[34], suggesting that BTN3A1 needs the support of other BTN3A members. Moreover, the association between BTN3A1 and BTN3A2, which occurs via their membrane-proximal IgC-like domains, was previously analyzed, and retention motif-dependent ER sequestration of BTN3A1 was shown to be rescued by co-expression of BTN3A1 with BTN3A2 and resulting BTN3A1-3A2 heteromer formation[33]. How this relates to increased or altered PAg sensing functionality remains unclear. Furthermore, the exchange of the JM of BTN3A1 for that of BTN3A3 increases this activation[36]. Nevertheless, how the BTN3A3 JM region contributes to enhanced function remains unknown.

Here, in order to define minimal requirements of the different BTN3A molecules for PAg-induced activation of Vγ9Vδ2 T cells, we express combinations of wild-type and mutated BTN3A molecules in BTN3A-deficient 293T (BTN3KO) cells and demonstrate that the functional features of various BTN3A molecules can be merged into a "super-BTN3" molecule, similar to a hypothesized primordial BTN3A present in species that encode single BTN3A isoforms such as alpaca[28,34,37]. We describe the BTN3A molecules as complexes in which a division of labor takes place that favors optimal T cell activation, whereby PAg sensing is initiated by the B30.2 domain of one BTN3A chain and an intact IgV domain must be present within the paired BTN3A chain of each dimer. Our results show that the BTN3 JM region controls both the trafficking and conformation of homomeric and heteromeric BTN3A complexes. In these complexes, the PAg-bound state is accompanied by the binding of the BTN3A1-B30.2-PAg complex to the B30.2 domain of BTN2A1. These results not only clarify the molecular mechanism underlying PAg-mediated activation of Vγ9Vδ2 T cells but also have implications for γδ T cell activation by other BTNs or butyrophilin-related molecules such as BTNL or SKINT family members[38].

## Results

### Loss of function of VΔ3A1 compensated in heteromeric BTN3A complexes

At first, we validated the necessity of all three isoforms for an optimal PAg response by testing the inactivation of different BTN3A genes in 293T cells (Fig. 1a–d)[33,34]. To this end, we employed the murine reporter TCR-transductant MOP 53/4 r/mCD28 cell line (TCR-MOP), which shows no cross or self-presentation as is observed for human γδ T cells[15,24,39]. The stimulation of the reporter TCR transductants is abrogated by BTN3A1 deficiency alone or by knockout of both BTN3A2 and BTN3A3, leaving behind the residual BTN3A1. The data revealed that BTN3A1 is the quintessential BTN3A molecule[34], and in its absence, BTN3A2 and BTN3A3 cannot elicit any PAg response. Second, BTN3A1 cannot elicit any response in the collective absence of BTN3A2 and BTN3A3, suggesting their essential contribution to a decent T cell activation under these conditions. Besides, a clear reduction in stimulation was observed for BTN3A2- or BTN3A3-deficiency, with BTN3A2 deficiency having a stronger impact. A similar outcome was observed with primary human Vγ9Vδ2 T cells as responders, except that the loss of BTN3A3 alone was not as impactful as seen with TCR transductants. We also demonstrated the cooperation of BTN3A isoforms by transduction with 3A1mC (3A1 mCherry fusion construct) alone or in combination with 3A2, 3A3 or 3A2 plus 3A3 in 293T cells with all three BTN3A genes inactivated (BTN3KO cell line or 3KO). Additionally, 3KO cells that expressed 3A2 or 3A3 in the absence of 3A1 did not result in activation. Notably, we did not test 3KO cells cotransduced with BTN3A2 and BTN3A3 because BTN3A1KO cells that express the endogenous BTN3A2 and BTN3A3 showed no T cell activation (Fig. 1a). Subsequently, all the experiments were performed in the 293T BTN3KO (3KO)[34] background and recombinant BTN3A derivatives were designated as 3A. A schematic overview of the constructs used in the study is provided in Fig. 1i.

The binding of BTN3A-V to the Vγ9Vδ2 TCR has been reported[40] but has not been confirmed by surface plasmon resonance[18], isothermal titration calorimetry[18] or by staining of BTN3A1 transductants with Vγ9Vδ2 TCR-tetramers[13]. To test the function of the human BTN3A family member V-domains, we generated recombinant BTN3A V-domain deletion mutants (VΔ) in which V domains were replaced by a FLAG-sequence preceded by a BTN3A1 leader sequence. If not explicitly stated, 293T BTN3KO cells (3KO)[34] were used as recipients for gene transduction. VΔ3A1 or VΔ3A2 were transduced alone or together with 3A1, 3A1mC, 3A2 or 3A3. VΔ3A3 was not tested since expression in 3KO cells failed. A sequence alignment of BTN3A molecules with relevant domains and regions marked is shown in Supplementary Fig. 1a. The transductants were sorted for similar BTN3A expression with the V-specific 103.2 mAb (Supplementary Fig. 1d, e) and stained for total expression (intracellular + surface expression of permeabilized and fixed cells) and surface expression (live cells) of the FLAG tag (Fig. 1e, g). Flow cytometry revealed that the VΔ3A1 transductant displayed no surface staining of the FLAG-tag unless a heterologous 3A-molecule was co-expressed (3A2 or 3A3 but not 3A1), and

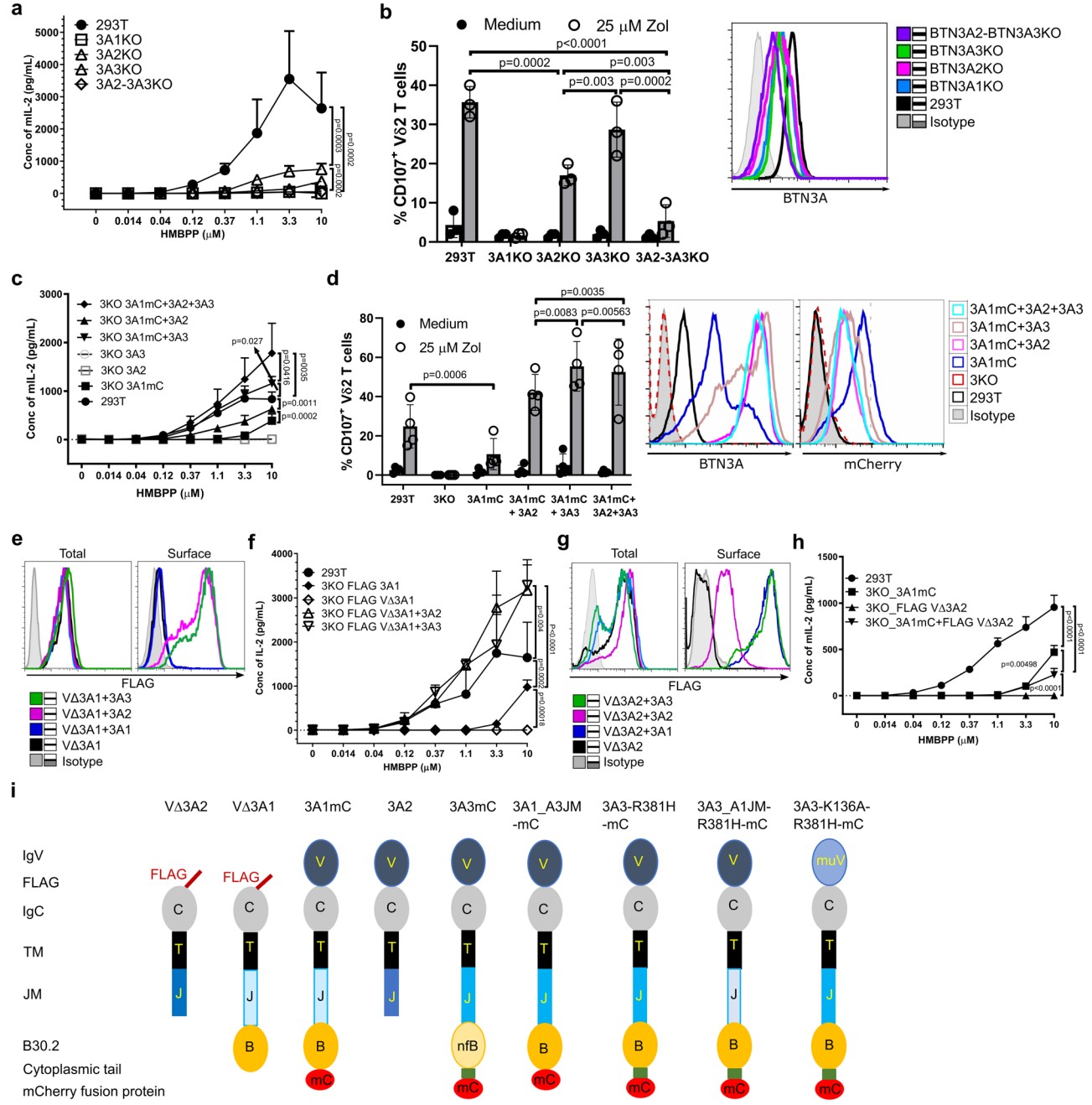

Color code for domains in 1i: FLAG tag - red stick; IgV (V) - navy blue; mutant-IgV (muV) - light blue; IgC (C) - gray; JM (J) of 3A1 - sky blue; 3A2 - dark blue, 3A3 - torquise blue; PAg sensing B30.2 (B) - yellow; PAg non-sensing B30.2 (nfB) - light yellow; mCherry fusion protein (mC)

**Fig. 1 | Loss of function of 3A1-V domain deleted molecules can be compensated in complexes with 3A2 or 3A3 molecules. a** 293T and BTN3 isoform-specific knockout cell lines were cocultured with titrated concentration of HMBPP and 53/4 human Vγ9Vδ2 TCR reporter cells. The activation of reporter cells was measured by mouse IL-2 ELISA (n-3). **b** The abovementioned presenting cells were pulsed with zoledronate and cocultured with HMBPP expanded primary Vγ9Vδ2T cells. The T cell activation was measured by immuno flow cytometry with CD107a expression as readout detected by anti-CD107a-PE and anti-Vδ2-FITC (n-3). Surface-expressed BTN3A of the abovementioned cells detected by mAb 103.2 followed by anti-mouse F(ab')2-APC conjugate (right). **c** 293T, BTN3KO (3KO) cells and 3A-transductants (3A represents recombinant BTN3A molecules) of 3KO were cultured and tested as in (**a**) (n-3). Not shown are the results of 293T 3KO as they are consistently non-stimulatory[34]. **d** Abovementioned presenting cells were tested as in (**b**) (n-4); their surface-expressed 3A-molecules were detected as in (**b**), and their corresponding

total mCherry expression was presented as histograms (right). **e** Histograms representing the total and surface-expressed FLAG protein of fix-permeabilized and live 3KO cells transduced with FLAG-tagged IgVdeleted-BTN3A1 (VΔ3A1) alone or cotransduced with other 3A-molecules detected by anti-FLAG and anti-mouse F(ab')2-APC conjugate were analyzed by FACS. **f** 3KO cells transduced with FLAG-3A1 or VΔ3A1, or VΔ3A1+3A2 or + 3A3 were cocultured with 53/4 Vγ9Vδ2 TCR reporter cells, and T cell activation was measured as in (**a**) (n-3). **g** 3KO cells expressing FLAG-IgVdeleted-BTN3A2 (VΔ3A2) alone or together with other BTN3As were analyzed as in (**e**). **h** 293T wt and 3KO cells transduced with 3A1 and/or VΔ3A2 were analyzed as in (**a**) (n-3). **i** Schematic representations of different tagged constructs of 3A and 3A mutants. The number of independent experiments was represented as *n*. Graphical data are presented as mean with SD, and statistical analysis was performed using ordinary two-way ANOVA analysis.

this result was confirmed with confocal microscopy (Supplementary Fig. 1f). Cell surface FLAG-staining of VΔ3A2 also required co-transduction of intact 3A-molecules. In this case, the reconstitution of FLAG-epitope surface expression by homologous 3A2 was weak but efficient for the heterologous 3A1 and 3A3 (Fig. 1g). In conclusion, the lack of the V-domain disrupts the BTN3A trafficking to the cell surface and staining of such VΔ-domain constructs (FLAG-VΔ3A) required co-expression of appropriate full-length BTN3A molecules.

Next, we tested for HMBPP-induced stimulation of the MOP TCR-transductant cell line[15,24,39]. 3KO cells transduced with VΔ3A1 and 3A2, or VΔ3A1 and 3A3 stimulated better than wild-type 293T cells, while cells co-expressing VΔ3A2 and 3A1 stimulated even worse than cells expressing only 3A1 (Fig. 1f, h). This reduced efficacy was not an effect of the FLAG-tag as tagged and 3A1 expressing 3KO cells showed similar responses (Supplementary Fig. 1c). Notably, protein domains contained in the complexes of VΔ3A1 and 3A2, or VΔ3A2 and 3A1, are identical (Fig. 1i), indicating that functional differences of the complexes result from the different localization of domains within the complexes, as will be discussed later.

### The JM region regulates BTN3A protein interaction and function

A major difference when comparing BTN3A1 relative to both BTN3A2 and BTN3A3 is their JM region (Supplementary Fig. 1a). To address its role in BTN3A isoform interaction and function, FLAG-VΔ3A1 was co-expressed with HA-tagged 3A1 or 3A1 containing the JM of 3A3 (3A1_A3JM). In cells with similar total levels (intracellular and cell surface) of FLAG-VΔ3A1, its surface expression was detected by flow cytometry only when cotransduced with HA-3A1_A3JM but not HA-3A1 (Fig. 2a). This finding suggests that the BTN3A1 JM region might hinder formation of fully functional BTN3A complexes while the heterologous BTN3A3 JM region may support such complexes. The ratio of cell surface to total expression was also considerably higher for HA-3A1_A3JM compared to wild-type HA-3A1 (Fig. 2a). This demonstrates the capacity of 3A3JM to alter the pattern of cellular distribution of 3A1_A3JM as well as the associated FLAG-VΔ3A1. Similar observations were made using confocal microscopic examination of immuno-stained live 3KO cells expressing FLAG-VΔ3A1 and HA-3A1 or HA-3A1_A3JM (Fig. 2b). Immuno-staining with anti-FLAG antibody detected the FLAG-VΔ3A1 (red) at the cell surface under live conditions only when cotransduced with HA-3A1_A3JM (right) but not with HA-3A1 (center). Furthermore, HA-3A1 or HA-3A1_A3JM (blue) proteins were clearly detected at the cell surface by anti-HA antibody, validating the presence of full-length proteins at the cell surface. Under fixed conditions, HA-3A1 was detected both in nuclear periplasmic space and at the plasma membrane, whereas HA-3A1_A3JM was detected predominantly at the plasma membrane and weakly in cytoplasmic vesicles. In 3KO FLAG-VΔ3A1 cells, FLAG-VΔ3A1 was detected only around the nuclear periplasmic space distinct from membranes labeled with BODIPY-FL (yellow). The detection of FLAG-VΔ3A1 around the nuclear periplasmic space remained the same when co-expressed with HA-3A1 or HA-3A1_A3JM (Fig. 2b, Fixed). However, FLAG-VΔ3A1 was also detected at the surface when co-expressed with HA-3A1_A3JM, exhibiting significant colocalization (violet), and this is not the case with HA-3A1. Notably, FLAG-VΔ3A1 colocalization with HA-3A1 was detected around the nuclear periplasmic space and a few vesicles (Fig. 2b and Supplementary Fig. 2a). Furthermore, similar observations were made with cells that expressed FLAG-VΔ3A1-CFP with 3A1-YFP or 3A1_A3JM-YFP (Supplementary Fig. 2c). Despite high expression of FLAG-VΔ3A1-CFP in cells, a little FLAG (yellow) was detected at the surface when co-expressed with 3A1_A3JM-YFP cells but not with 3A1-YFP. Finally, a microscopic examination of these cells revealed the altered trafficking of 3A1_A3JM attributed to 3A3JM.

We performed immunoprecipitations (IP) using the cells mentioned above to extend our findings to biochemical interactions. Cell lysates were subjected to anti-FLAG IP and subsequent anti-HA Western blot (Fig. 2c). In line with the colocalization of FLAG-VΔ3A1 with HA-3A1 under fixed-permeabilized conditions and at the cell surface for FLAG-VΔ3A1 with HA-3A1_3A3M, IP demonstrated potential interactions between FLAG-VΔ3A1 with HA-3A1 or HA-3A1_A3JM but did not show any differences in the quantities of co-precipitated HA-proteins. The differential size of HA-3A1_A3JM and HA-3A1 in the immunoblot coincided with their differential localization and trafficking.

### BTN3A3 JM promotes close association of B30.2 domains in BTN3A complexes

Although the VΔ3A1 association was observed with both HA-3A1 and HA-3A1_A3JM constructs in IP, the differential surface expression of VΔ3A1 led us to postulate that the resulting heteromeric 3A complexes adopted different conformations. FRET analysis was used to test the interaction between fluorescent fusion proteins and to infer the conformation or mode of association between 3A molecules within homomers or heteromers. For FRET assays, 3KO co-transductants of FLAG-VΔ3A1-CFP or FLAG-3A1-CFP and 3A1-YFP or 3A1_A3JM-YFP were generated (Fig. 2d). FRET ratio was measured as stipulated in the "Methods" section and acquired images are presented as ratiometric images (Fig. 2e).

The setup was optimized with 3KO single transductants of FLAG-3A1-CFP and 3A1-YFP/3A1_A3JM-YFP constructs; the intensity 480/30 and 535/40 filters were similar with CFP constructs, and no image was visualized with YFP constructs as YFP was not excited by a 440 nM CoolLED (Supplementary Fig. 2d).

Neither full-length 3A1-CFP nor VΔ3A1-CFP co-expressed with 3A1-YFP displayed any FRET (Fig. 2e, left panel), and they yielded images with similar intensities with both the filters, suggesting for this experimental setup no FRET between CFP and YFP (B30.2 domains) either on the plasma membrane or in the cytoplasmic compartments (Supplementary Fig. 2d). On the other hand, 3A1-CFP co-expressed with 3A1_A3JM-YFP revealed high FRET predominantly at the plasma membrane (Fig. 2e, upper right), and with the increased intensity with the 530-nM filter (Supplementary Fig. 2d).

Even stronger FRET was observed at the membrane when FLAG-VΔ3A1-CFP was co-expressed with 3A1_A3JM-YFP (Fig. 2e, lower right). This was consistent with observations from immune staining and confocal microscopy (Fig. 2b and Supplementary Fig. 2c), where 3A1_A3JM was overwhelmingly detected at the plasma membrane but not in cytoplasmic organelles, and in spite of the predominant cellular retention of the VΔ3A1 protein, detectable levels of FLAG-tagged protein managed to reach the plasma membrane when cotransduced with 3A1_A3JM. Furthermore, analysis of the total FRET (cytoplasmic and membrane) also demonstrated a noteworthy FRET signal in 3KO cells co-expressed FLAG-VΔ3A1-CFP, or FLAG-3A1-CFP with 3A1_A3JM-YFP but not with 3A1-YFP (Supplementary Fig. 2e).

Collectively, these data suggest that expression of FLAG-VΔ3A1-CFP or 3A1-CFP with 3A1-YFP led to 3A-complexes where B30.2 domains are distantly spaced. On the contrary, co-expression of FLAG-VΔ3A1-CFP or 3A1-CFP with 3A1_A3JM suggests the formation of 3A-complexes in which their respective B30.2 domains are in FRET-able distance.

### A division of labor in BTN3A heteromers and super-BTN3 homomers

Alpaca-like species demonstrating single BTN3-dependent PAg responses led us to postulate a single BTN3 molecule as a primordial requirement, and it was of interest to generate such a BTN3 protein, which encompasses requisite domains for the PAg-dependent response. To this end, 3KO cells transduced with mCherry (mC) fused to 3A1 (3KO_3A1mC), 3A3 gain of function mutant R381H (3KO_3A3-R381H-mC), 3A1 with the JM of BTN3A3 (3KO_3A1_A3JM-mC) and finally with a gain of function 3A3 mutant possessing JM of 3A1 (3A3_A1JM_R381H-mC) were analyzed (Fig. 3a–d). In the functional

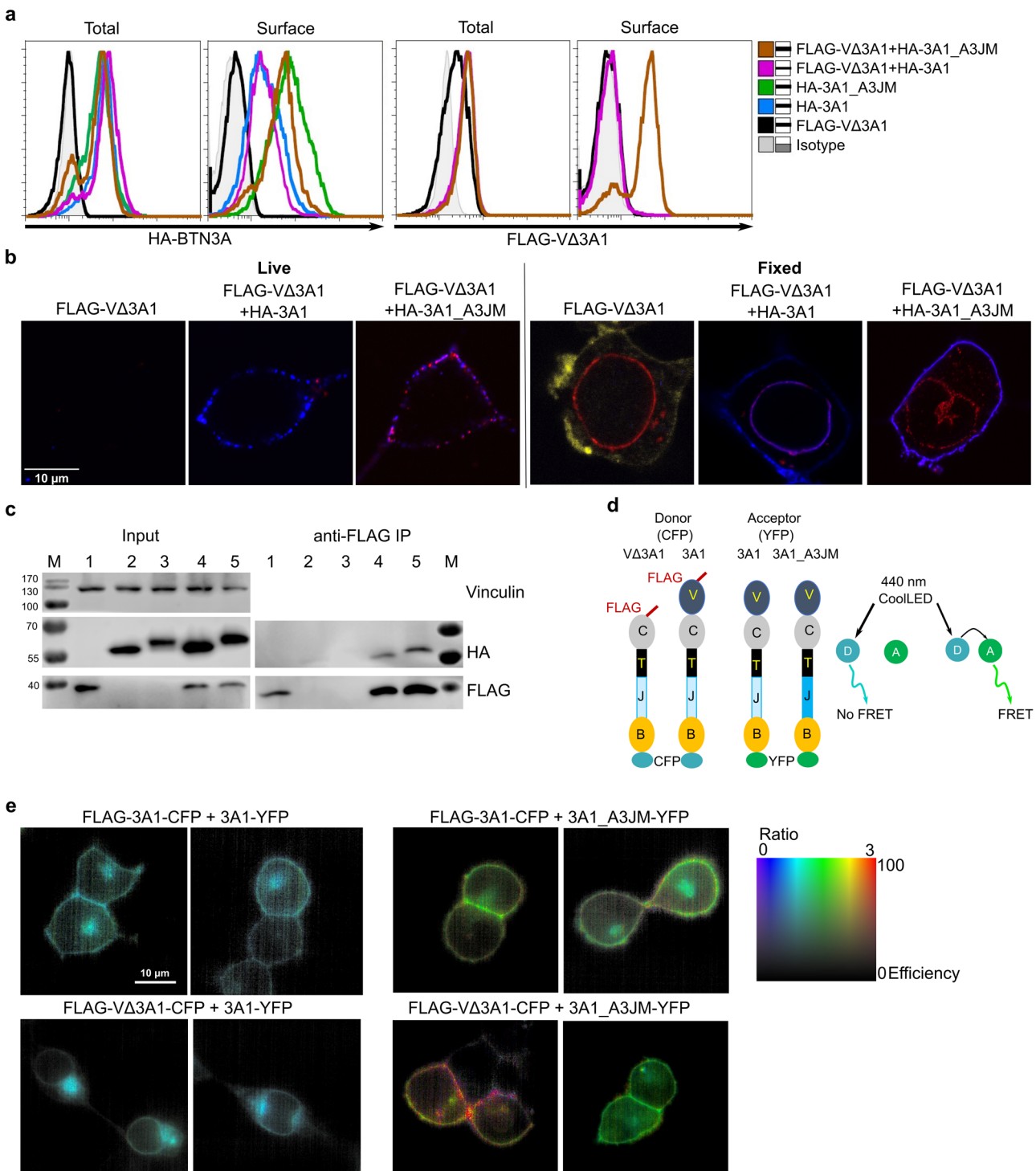

Color code for domains in 2**d**: FLAG-tag - red stick; IgV (V) - navy blue; mutant-IgV (muV) - light blue; IgC (C) - gray; 3A1JM (J) - sky blue; 3A3JM (J) - torquise blue; PAg sensing B30.2 (B) - yellow; cyan fluorescence protein (CFP) - cyan; yellow fluorescence protein (YFP) - green

assay (Fig. 3a), cells expressing 3A-proteins with a functional PAg sensing B30.2 domain and the 3A3 JM region were indistinguishable from 293T cells, whereas cells expressing 3A1-mC or 3A3_A1JM_R381H-mC that possess the 3A1 JM region were very poor stimulators, and as expected 3A3 expressing cells did not stimulate at all. Analysis of recombinant BTN3A protein distribution in these cells revealed that despite a similar degree of mCherry fusion protein expression (Fig. 3b), the cells exhibited pronounced differences in intracellular localization and in the formation of mCherry aggregates (Fig. 3c). In all cases, cells expressing 3A-molecules bearing exclusively 3A1JM

displayed a higher degree of intracellular retention of fluorescent complexes than their 3A3JM expressing counterparts, which displayed enhanced expression at the plasma membrane (Fig. 3c). As a positive correlation has been reported for immobilization of surface BTN3A molecules in the presence of stimulants like aminobisphosphonates (pamidronate) or the agonist mAb 20.1 with the capacity of T cell activation[15,18], finally, we tested the effects of such stimulants on the cell surface immobility of 3A-molecules by FRAP (Fluorescence Recovery after Photobleaching)[15]. Constructs with a 3A1JM displayed no increased immobilization, whereas those with a 3A3JM did (Fig. 3d).

**Fig. 2 | The JM region regulates BTN3A protein and function. a** 293T 3KO cells transduced with FLAG-VΔ3A1 alone and or cotransduced with N-terminus HA-tagged 3A-JM chimeras were analyzed in FACS for the total and surface expression of HA-3A molecules (Left) and FLAG-VΔ3A1 (right). The measurements were presented as histograms. **b** A representative image of live (left) and fixed (right) 3KO cells transduced with FLAG VΔ3A1, cotransduced with HA-3A1 or HA-3A1_A3JM chimera, that were stained with mouse anti-FLAG and rabbit anti-HA followed by anti-mouse-Alexa Fluor 647 (red) and anti-rabbit Alexa Fluor 555 (blue), respectively. 3KO-FLAG VΔ3A1 cells additionally stained with BODIPY-FL-DHPE membrane dye (yellow). At least 6 images of 3KO cells expressing each construct were examined under live and fixed conditions. **c** 3KO cells transduced with FLAG-VΔ3A1, HA-3A1, HA-3A1_A3JM, FLAG-VΔ3A1+HA-3A1, and FLAG-VΔ3A1+HA-3A1_A3JM were labeled as 1–5, were subjected to anti-FLAG immunoprecipitation (IP) and samples were blotted against human vinculin (input, top), FLAG (middle) and HA (bottom) for their input (left) and immunoprecipitated proteins (right) (n=2). **d** Schematic presentation of FLAG-VΔ3A1-CFP, FLAG-3A1-CFP, 3A1-YFP and 3A1_A3JM-YFP constructs (left), scheme describing the FRET with 440 LED laser, D is the donor (CFP), A is the acceptor (YFP) and A will emit a signal when exited by D if it is close proximity showing FRET. **e** Two representative images for ratiometric FRET analysis of 3KO transduced with 3A1-YFP and FLAG-3A1-CFP (upper left) or FLAG-VΔ3A1-CFP (lower left); 3KO transduced with 3A1_A3JM-YFP and FLAG-3A1-CFP (upper middle) or FLAG-VΔ3A1-CFP (lower middle); FRET ratio (FR) calculated chart (right).

Notably, medium controls of the cells expressing the 3A3JM-containing constructs also displayed a higher degree of immobilization than that of the transductants with 3A1JM-containing constructs (3A1mC and 3A3-A1JM-R381H-mC), which is consistent with the reported higher background stimulation for activation of short term Vγ9V δ2 T cell lines by 293T transfected with 3A1_A3JM[36] or 3A3_A1_B30.2 and 3A3_R381H[18]. Likewise, cells expressing 3A1-mC plus 3A2-3A3 (Supplementary Fig. 3a) behaved analogously to cells expressing the 3A3 JM-containing constructs in terms of intracellular trafficking and aggregate formation. Furthermore, native gel electrophoresis of solubilized membrane extracts revealed very large 3A1-mC complexes when prepared with detergent Brij 96 and Triton X100 (Supplementary Fig. 3b). In contrast, membranes solubilized with digitonin, which binds to cholesterol, massively reduced the size of 3A1mC molecular complexes (>440 kDa). In the presence of 3A2 and 3A3, these complexes were dissociated into two complexes of less than 440 kDa apparent Molecular mass (Mr)[18]. These observations suggest that huge complexes formed by 3A1 in the absence of other BTN3As may correlate with poor trafficking to the membrane and dominant retention in ER vesicles due to their ER retention motif[33]. In contrast, the co-expression of 3A2 and 3A3 with 3A1 resulted in relatively smaller complexes that coincided with improved trafficking of BTN3A complexes to the membrane (Supplementary Fig. 3a). Altogether, the 3A3-JM-containing constructs can substitute for "help" for 3A1 JM by 3A2 or 3A3 in terms of stimulation capacity, cellular trafficking of 3A proteins, and formation of molecular clusters.

So far, we showed that altered functional properties such as poor trafficking and immobilization of BTN3A complexes comprising exclusively BTN3A1JM coincided with reduced stimulatory capacity, unlike heterologous BTN3A complexes incorporating BTN3A3JM and exhibiting an optimal PAg response. Surprisingly, VΔ3A1+3A2 and VΔ3A2+3A1 complexes stimulated quite differently, although the surface expression of BTN3A molecules was similar (Fig. 1f–h). Moreover, as depicted in Fig. 3g, both complexes possess sequence-identical protein domains and differ only in the relative arrangement of the V domains. In one case, the IgV domain is located on the PAg-binding protein (3A1) and in the other on the pairing chain (3A2). This feature relates back to a previous report on V-domain mutants (K136A) affecting PAg-mediated stimulation[41], where complexes of 3A2_K136A and 3A1 lost stimulatory potential while complexes of 3A1_K136A and 3A2 did not. To test whether similar effects were also observed for a homomeric "super" BTN3A" (3A3_R381H), a mutant with a substitution at position 136 was generated (3A3_R381H_K136A-mC). 3A3_R381H_K136A-mC was co-expressed with one of two different PAg-binding-insufficient BTN3A-IRES-GFP reporter constructs (3A3 (GFP) or 3A1_H381R(GFP)) and were sorted for similar surface BTN3 expression and their corresponding fluorescent reporter (GFP) (Fig. 3e). As expected, cells transduced with V-domain mutant (3A3_R381H_K136A-mC), failed to stimulate the TCR transductants. Furthermore, stimulation was successfully detected with both the co-transductants whose BTN3A complexes are composed of a PAg-sensing molecule (V-domain mutant) and a wild-type V-domain-containing molecule (PAg-binding

insufficient). This implies a distinct topology of BTN3A complexes is required for successful PAg sensing, whereby the PAg-binding site and wild-type V-domain are located on different molecules (Fig. 3f, g). This is consistent with the differential stimulatory capacity of VΔ3A1+3A2 vs VΔ3A2+3A1 transduced cells (Fig. 1f, h). Altogether, these results suggest that PAg binds to one BTN3A molecule that, via the JM region, is connected to a paired BTN3A molecule whose intact V-domain is essential for PAg sensing mediated via the Vγ9V δ2 TCR.

## A structural rationale for heteromeric BTN3A coiled-coil assembly

To probe the differential impact of the JM region on BTN3A function, we compared the sequence of BTN3A1JM to that of other BTN3A molecules (Supplementary Fig. 1a and Fig. 4a). We noted that the JM of BTN3A1 contains a positively charged motif (R**KKK**R) (position 283-285) while BTN3A2, BTN3A3, and alpaca BTN3A possess two negatively charged glutamic acid residues (xExEx) at this position (Fig. 4a). Taking into account the well-established importance of electrostatic interhelical interactions in determining the stability of coiled-coil domains[42,43] we speculated that differences within this strong positively charged motif may disfavor homodimerization of BTN3A1 and instead favor BTN3 heterodimerization. To test this, we designed mutants to swap merely the KKK and ETE of BTN3A1 and BTN3A3 to determine their contributions to both BTN3A association and their impact on the functional efficacy of the BTN3A molecule in the PAg response. As expected, the substitution of the BTN3A3 ETE motif by KKK (3A3-KKK) abolished the rescue of surface expression of FLAG-VΔ3A1 and reduced the stimulatory activity to that of 3A3-R381H-KKK-mC (Fig. 4b, c). This suggested that this triplet motif is essential for the JM-mediated interaction of 3A1 and 3A3 molecules. Interestingly, the replacement of KKK of BTN3A1 by ETE (3A1_ETE) did not rescue FLAG-VΔ3A1 cell surface expression and did not change the stimulatory capacity of the 3A1, suggesting other regions of the JM may also be involved in controlling cooperation and trafficking of associated 3A-molecules (Supplementary Fig. 4a, b).

To probe the molecular basis of these effects, we carried out molecular modeling of the coiled-coil region of the BTN3A isoforms. We restricted these efforts to the 273–312 region that was previously strongly predicted to form a coiled-coil domain by mediating BTN3A dimer interactions[44], within which the BTN3A1 KKK 'triplet region' is located (283–285), and employed a parametric α-helical coiled-coil prediction methodology (CCBuilder 2.0)[45].

These efforts first highlighted the potential of human BTN3A1, BTN3A2, BTN3A3, and also the single alpaca isoform VpBTN3 to each form biophysically plausible homodimers via intermolecular coiled-coil interactions, stabilized in each case by numerous polar and non-polar interactions at the interhelical molecular interface. Of note, these models predicted interhelical interactions mediated by the 283–285 triplet residues that could partly account for differential stability and conformation (Fig. 4d) and therefore surface expression and functionality (Fig. 4b, c). In the BTN3A3 homodimer, E283 and T284 were predicted to form stabilizing hydrogen-bonding interactions to

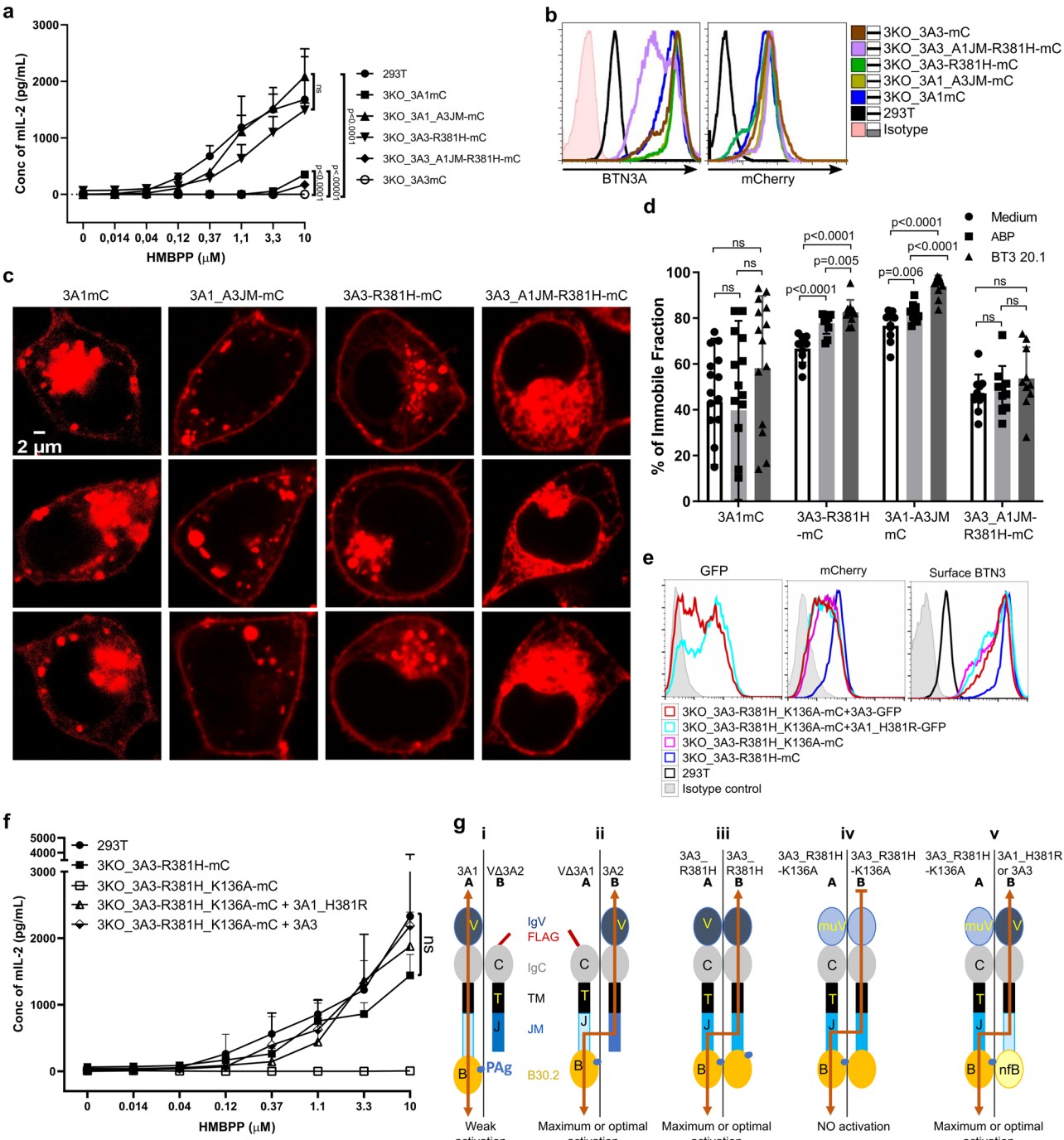

Color code for domains in 3**g**: FLAG tag - red stick; IgV (V) - navy blue; mutant-IgV (muV) - light blue; IgC (C) - gray; JM (J) of 3A1 - sky blue; 3A2 - dark blue, 3A3 - torquise blue; PAg sensing B30.2 (B) - yellow; PAg non-sensing B30.2 (nfB) - light yellow;
Brown line representing signal outcome, N-terminus arrow mark style represents activating complex, N-terminus T style represents non-activating complex

equivalent residues of the opposing helix, with the involvement of R288 from each monomer; in contrast, E285 was solvent-exposed and not involved in interhelical contacts (Fig. 4d I). In BTN3A2, I284 was the sole mediator of interhelical triplet region interactions comprised of non-polar interface contacts with the corresponding residue of the opposing helix (Fig. 4d II); unlike BTN3A3, E283 and E285 were solvent-exposed and uninvolved in intermolecular contacts. While biophysically feasible, the relative stability of this arrangement was unclear. Nevertheless, it is consistent with the weaker surface expression of VΔ3A2 when co-expressed with 3A2 compared to that of co-expression with 3A1 and 3A3. Similar to human BTN3A3, modeling of the single alpaca-encoded 'superagonist' isoform, VpBTN3, indicated

involvement at the interhelical interface of E283 and K284, which mediated reciprocal salt-bridge interactions with the same pair of residues from the opposing monomer (Fig. 4d III). Notably, for the BTN3A1 model, the indicated 'KKK' at the 283-285 region was arranged differently, with 284 and 285 positioned at the interhelical interface and 283 solvent-exposed and uninvolved (Fig. 4d IV). Most importantly, this model predicted the positively charged K284 and K285 were directly facing the same residues from the opposing monomer at the interface (Fig. 4d IV). This arrangement is likely to be energetically highly unfavorable and destabilize the BTN3A1 homodimer via electrostatic repulsion; moreover, consistent with results from FRET analyses (Fig. 2), it may favor a weaker intermolecular association.

**Fig. 3 | Homologous 3A3_JM and heterologous 3A_JM promote optimal stimulation via inter-BTN3 PAg signaling. a** 293T and 3KO transductants of 3A-constructs were cultured with HMBPP and 53/4 human Vγ9Vδ2 TCR reporter cells. The activation of reporter cells was measured by mouse IL-2 ELISA (n-3). **b** Surface-expressed 3A-proteins of the abovementioned cells detected by mAb 103.2 followed by anti-mouse F(ab')2-APC conjugate (left) and their corresponding total mCherry expression (right) were presented as histograms. **c** The cellular distribution of BTN3A-mC fusion constructs is presented as images captured by confocal microscopy. At least 10 images of 3KO cells expressing each construct were analyzed, and 3 representative images of 3KO cells expressing each construct are shown here. **d** mCherry fusion constructs of 3A or 3A-JM chimera transduced 3KO cells were subjected to FRAP in the presence of pamidronate (250 μM) and mAb 20.1 (10 μg/mL) as previously reported[15], and the percentage of the immobile fraction of BTN3A-mC was measured. The number of cells (*n*) subjected to FRAP for 3KO_3A1mC (n-15) and other cell types (n-10) for each condition. **e** 293T, 3KO

transduced with mCherry fusion constructs of 3A3_R381H, 3A3_K136A_R381H, and cotransduced with eGFP reporter constructs of 3A1_H381R or 3A3 were analyzed by FACS for their total mCherry, total GFP, and surface-expressed BTN3As shown as histograms as in (**b**) (bottom right). **f** The abovementioned cells were tested as in (**a**) (n-3). The inferred intermolecular signaling within the BTN3A proteins viz 3A3_R381H, 3A3-K136A-R381H, and 3A3/3A1_H381R and the observed stimulation strength was presented as a scheme in (**g**) iii, iv and v, respectively. **g** Schematic presentation of inferred intermolecular signaling within the BTN3A proteins correlated to the observed outcomes in terms of 53/4 human Vγ9Vδ2 TCR reporter activation strength with antigen-presenting cells (3KO) expressing VΔ3A2 and 3A1 (i, represents Fig. 1h), VΔ3A1 and 3A2 (ii, represents Fig. 1f) including the 3A-constructs mentioned in (**f**); Graphical data are presented as mean with SD were analyzed by ordinary two-way ANOVA (**a**, **f**) or multiple *t*-tests analysis (**d**) with statistical significance determined using the Bonferroni-Dunn method and SD was shown as error bars.

Therefore, while biophysically feasible, BTN3A1 modeling highlights the KKK motif of BTN3A1 is likely to disfavor homodimer formation in a way that is not predicted to occur with other isoforms.

Modeling approaches also shed light on heteromeric interactions. BTN3A1/3A2 (Fig. 4d V) and BTN3A1/3A3 (Fig. 4d VI) coiled-coil models highlighted not only a loss of the interhelical electrostatic repulsion evident from the 283–285 region of BTN3A1 homodimers (Fig. 4d IV) but also predicted a favorable salt-bridge interaction from K285 of 3A1 to E283 of BTN3A2/3A3. This was consistent with more stable coiled-coil heterodimers relative to the BTN3A1 homodimer, including a potential for closer intermolecular association between the two BTN3A chains in this context, consistent with the results of the FRET analyses. Of note, modeling of BTN3A3 mutated to incorporate the KKK motif of BTN3A1 at 283-285 (Fig. 4d VII) indicated the close opposition of K283 and K284 to identical residues across the interhelical interface. Although this differed from the predicted native BTN3A1 dimer interface, where K284 and K285 are localized to the dimer interface, it was nevertheless likely to substantially destabilize the BTN3A3-KKK dimer and was entirely consistent with the pronounced deleterious effect of the BTN3A1 JM region (Figs. 1 and 3) and KKK motif (Fig. 4) on both surface expression, conformation, and functionality.

Finally, inspection of the models strongly indicated that extra-triplet effects contribute to differential homodimer and heterodimer stability (Supplementary Fig. 4, Supplementary text). In particular, the 276–278 region appeared particularly significant (Supplementary Fig. 4c I–VI), as it was predicted to form stabilizing non-polar (BTN3A2 homodimers) (Supplementary Fig. 4c II) or salt-bridge interactions (BTN3A3 homodimer, alpaca BTN3 homodimer, BTN3A1/A2 heterodimer, BTN3A1/A3 heterodimer) (Supplementary Fig. 4c III-VI), whereas in BTN3A1 the presence of K277 and K278 introduced electrostatic repulsion at the dimer interface (Supplementary Fig. 4c I). Moreover, the intermolecular packing interactions mediated by L280 in all other isoforms were lost in BTN3A1 homodimers (Supplementary Fig. 4c VII–X), in which the polar residue (Q) at this position was predicted to be solvent-exposed (Supplementary Fig. 4c VII). In summary, modeling studies suggest that interhelical interactions outside of the 283–285 region preferentially destabilize BTN3A1 homomers relative to both BTN3A2/3 homomers and also relative to heteromers involving BTN3A1 and BTN3A2/A3. This provides a potential molecular explanation for the observation that the introduction of the 283–285 ETE sequence of BTN3A3 into 3A1 is insufficient to confer substantially increased expression and functionality (Supplementary Fig. 4a, b).

**4-M-HMBPP disrupts the interaction of BTN3A1-BTN2A1 B30.2 domains**

We next compared HMBPP and 4-M-HMBPP, a synthetic HMBPP derivative incorporating a bulky head group that permits HMBPP-like binding to the BTN3A1-B30.2 domain but has a massively reduced stimulatory capacity compared to HMBPP that has been suggested to

result from an "aberrant" BTN3A1-B30.2 homodimer preventing BTN3A1 from adopting a hypothesized stimulatory conformation[46]. We previously demonstrated that the intracellular domains of BTN2A1 and BTN3A1 interact, but only in the presence of a potent PAg such as HMBPP[20]. Here we examined the ability of 4-M-HMBPP to support this interaction by using isothermal titration calorimetry (ITC). We confirmed a robust binding interaction between 4-M-HMBPP and the BTN3A1 full intracellular domain (3A1 BFI), which is a purified protein composed of the JM region and B30.2 domain of BTN3A1 (Fig. 5c), albeit with lower binding affinity of 2.9 μM than reported in ref. [47] that may result from different 3A1 constructs or compound purities. Next, we titrated the BTN2A1 intracellular domain (2A1 ID271, comprising the JM region and B30.2 domain) into 3A1 BFI. In agreement with our prior study, no interaction was observed in the absence of PAg (Fig. 5d), whereas in the presence of HMBPP, a strong interaction was observed ($K_D$, 0.8 μM) (Fig. 5e), which coincides with recently reported findings[21]. However, in the presence of 4-M-HMBPP, no binding occurred between BTN2A1 ID271 and BTN3A1 BFI (Fig. 5f), as shown in Table 1. Therefore, we can conclude that while 4-M-HMBPP binds to BTN3A1, it does not allow it to engage subsequently with BTN2A1. Together, the binding of PAg to BTN3A1 in the BTN3A heteromer allows it to interact with BTN2A1 homodimer to promote T cell activation.

## Discussion

This study addresses the contribution of BTN3A protein domains and their binding partners to PAg-induced Vγ9Vδ2 T cell activation. First, it demonstrates a crucial role for the V-domain, which is a prerequisite for cell surface expression of BTN3A chains. Second, the impaired trafficking of BTN3 lacking its membrane distal IgV-domain could be rescued by partnering preferentially with BTN3 molecules possessing the equivalent domain. Third, the functional contribution of the BTN3A membrane distal IgV domain to PAg stimulation can be compensated by the paired BTN3A molecule. Such compensation of loss of function BTN3A1-V constructs by residual levels of BTN3A2 and BTN3A3 isoforms could explain the observation that BTN3A1 V-domain mutants expressed in BTN3A1-knockdown 293T cells did not display any phenotype[48] while in HeLa cells knockdown of BTN3A2 and to a lesser extent of BTN3A3 reduced the HMBPP response[32]. It may also explain the reason behind the lack of response of the human Vγ9Vδ2 TCR transductant (TCR-MOP) to HMBPP-treated 3KO cells transduced with a chimera composed of alpaca BTN3 (V-C domain) and human 3A1 (transmembrane-JM-B30.2 domain) and the fact that TCR-MOP gains responsiveness when the same construct was transduced into BTN3A1KO cells. This suggests that in the latter scenario, chimera comprising BTN3A complexes involving V domains of endogenous BTN3A2 and/or BTN3A3 may engage with the human TCR upon PAg sensing by B30.2 domain of the chimera or permit its ligation by an associated ligand[34,37].

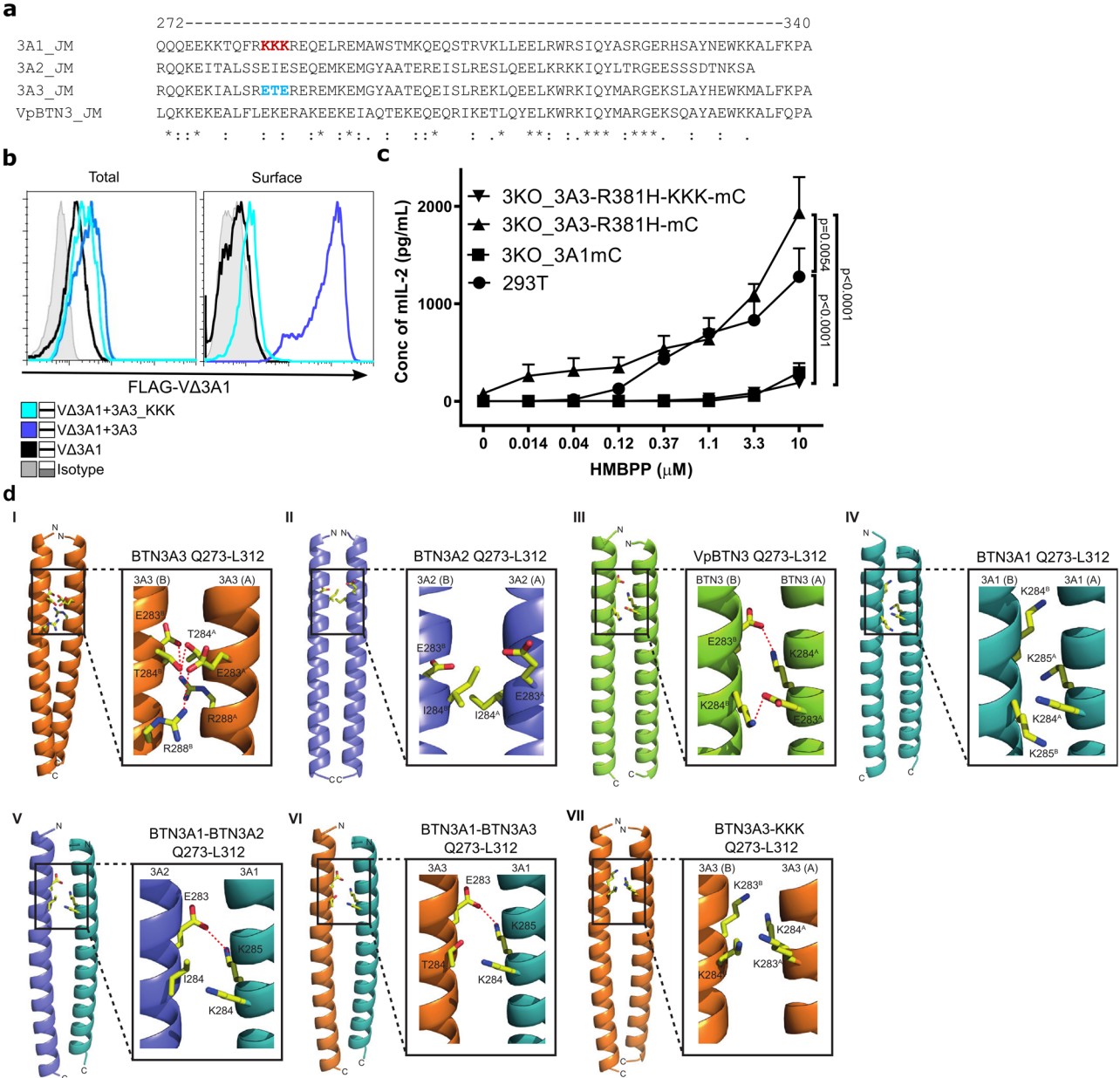

**Fig. 4 | JM regions modulate the conformation of BTN3A dimers. a** Amino acid sequences of juxtamembrane (JM) region of BTN3A1, BTN3A2, BTN3A3, and alpaca BTN3 (Vp) were aligned, and KKK and ETE residues of BTN3A1 and BTN3A3 were marked in red and blue, respectively. **b** Total and surface-expressed FLAG protein of permeabilized and live 3KO cells transduced with FLAG VΔ3A1 alone or cotransduced with 3A3 or 3A3_KKK mutant detected by anti-FLAG and anti-mouse F(ab')2-APC conjugate were shown as histograms. **c** 3KO cells transduced with 3A1mC, 3A3_R381H-mC, or 3A3_R381H_KKK-mC mutant were cocultured with 53/4 Vγ9Vδ2 TCR reporter cells and titrated concentration of HMBPP. The activation of reporter cells was measured by mouse IL-2 ELISA (n-3). **d** Models of the BTN3-JM coiled-coil dimers. Models of the predicted JM coiled-coil dimers Q273–L312 were generated using CCBuilder2 (see "Methods"). Dimer interface residues at positions 283–285 are shown as ball and stick. (I) BTN3A3 coiled-coil homodimer, (II) BTN3A2 coiled-coil homodimer, (III) Alpaca BTN3 (VpBTN3) coiled-coil homodimer, (IV) BTN3A1 coiled-coil homodimer, (V) BTN3A1-BTN3A2 coiled-coil heterodimer, (VI) BTN3A1-BTN3A3 coiled-coil heterodimer, (VII) BTN3A3-KKK (replacing ETE with KKK at positions 283–285) coiled-coil homodimer. Polar interactions are highlighted (red dashed lines). Each monomer within the homodimer has been labeled A or B. Graphical data are presented as mean with SD and analyzed by ordinary two-way ANOVA and SD was shown as error bars.

BTN3A2 as well as BTN3A3 reconstituted surface expression of VΔ3A1 and the resulting complexes permitted PAg-induced Vγ9Vδ2 TCR-mediated activation as efficiently as naturally occurring BTN3A heteromers or "super" BTN3As. In striking contrast, simultaneous expression of VΔ3A2 with BTN3A1, despite rescuing VΔ3A2 cell surface expression, failed to increase BTN3A1 mediated stimulation. Since the protein domains of surface-expressed 3A1-VΔ3A2 complexes and of 3A2-VΔ3A1 are identical, we conclude that localization of the V-domain within the complex is crucial for HMBPP-mediated stimulation. Such a topological effect could also explain the differential stimulation by 3KO cells co-expressing V-domain mutated BTN3A1 and wild-type BTN3A2 versus cells expressing wild-type BTN3A1 and mutated BTN3A2[41].

It is further supported by the HMBPP-induced stimulation by 3KO cells expressing homomer-like BTN3A3-derivatives consisting of 3A3 and V-domain mutated super BTN3 (3A3+3A3_K136A_R381H) whose possible mechanistic basis will be discussed later.

Several aspects of the contribution of the JM of BTN3A to PAg stimulation were analyzed in previous studies. First, PAg binding to the

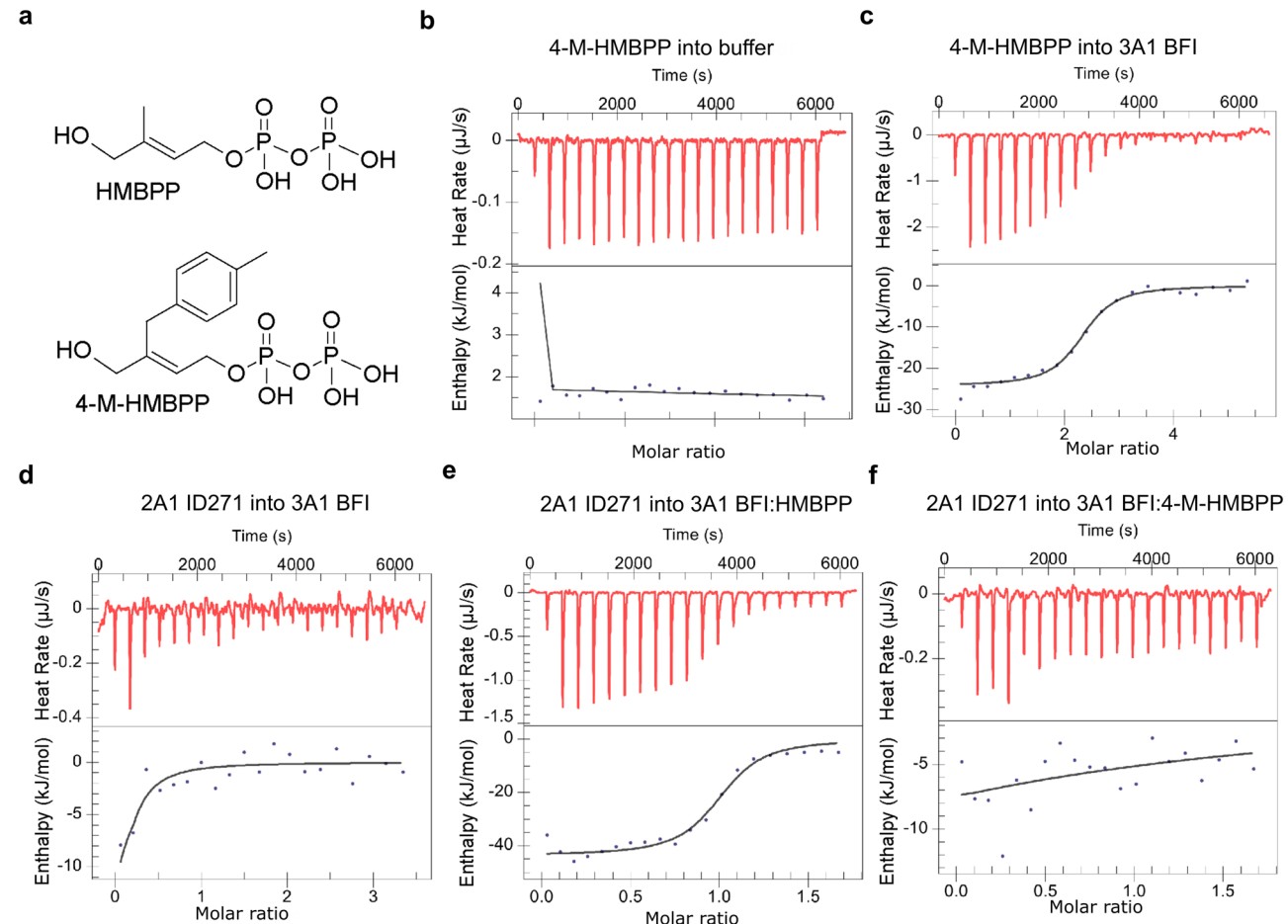

**Fig. 5 | 4-M-HMBPP-bound BTN3A1 did not interact with the BTN2A1-B30.2 domain.** ITC titrations show that 4-M-HMBPP binds to BTN3A1 but does not support the binding of BTN3A1 to BTN2A1. **a** Structure of HMBPP and 4-M-HMBPP. **b** Titration of 960 μM 4-M-HMBPP into the buffer. **c** Titration of 960 μM 4-M-HMBPP into 60 μM BTN3A1 BFI. **d** Titration of 600 μM BTN2A1 ID271 into 60 μM BTN3A1 BFI. **e** Titration of 300 μM BTN2A1 ID271 into a mixture of 60 μM BTN3A1 BFI and 120 μM HMBPP. **f** Titration of 300 μM BTN2A1 ID271 into a mixture of 60 μM BTN3A1 BFI and 120 μM 4-M-HMBPP. Results are representative of n-3 independent experiments; 3A1 BFI−BTN3A1 intracellular domain (JM+B30.2 domain); 2A1 ID271−BTN2A1 intracellular domain (JM+B30.2 domain).

B30.2 domain was described, and changes in the JM were found to be linked to PAg-induced stimulation[29,30]. Vantourout and colleagues noted the importance of the association of BTN3A1 and BTN3A2 molecules as well as the superiority of BTN3A1-BTN3A2 heteromers over BTN3A1 homomers in stimulation. Also identified were ER retention motifs in the JM of both molecules, which control intracellular trafficking and cell surface expression and are crucial for PAg-induced stimulation but could not explain the superiority of BTN3A heteromers over homomers[33]. Finally, the Scotet group showed an increase in stimulation after replacing the JM of BTN3A1 with 3A3JM[36]. Importantly, the current study can discriminate BTN3A complexes efficiently mediating PAg-stimulation from weak or non-stimulatory forms. It defines JM-controlled features: first, the rescue of surface expression of a paired V-deleted BTN3A molecule and second, in the case of BTN3A complexes, adaptation of a conformation that supports FRET between C-terminal fluorochromes. Notably, both cell surface rescue and efficient C-terminal FRET were not achieved for exclusively 3A1_JM-containing molecules unless they were co-expressed with other BTN3A2 or BTN3A3 or 3A3_JM-containing constructs. The high efficacy of heteromers that contain only a single PAg-binding site or, in the case of BTN3A1-BTN3A2 dimers, even only a single B30.2 domain over BTN3A1 homodimers is of special importance when discussing models postulating certain conformers of the extracellular domains (e.g., head to tail versus V-shaped dimers) or B30.2 domain dimers (symmetric versus asymmetric)[29,46,47,49,50] as being crucial for PAg-induced

activation. Intriguingly, the rescue of surface expression of VΔBTN3A1 as an indicator for the successful formation of BTN3A-complexes coincided very well with molecular modeling of coiled-coil structures formed by JM α-helices, which suggests reduced stability of 3A1 JM homodimers relative to BTN3A3 JM or alpaca BTN3JM homodimers and favor heteromeric BTN3A1 JM interactions with BTN3A2 or BTN3A3JM. The residual activation seen with (overexpressed) BTN3A1 or 3A1JM-containing constructs (Figs. 1a−d and 3a) might result from a small number of molecules still adopting a suitable extracellular BTN3A1-BTN2A1 topology despite unfavorable JM association[29,33,34]. Importantly, the significance of the BTN3 RKKKR/xExEx motif in affecting such preferences is entirely consistent with previous studies on coiled coils[44], which emphasize the significance of interhelical electrostatic interactions in dictating preference for homo-or heterodimer pairing in both native and de novo designed coiled coils, with oppositely charged pairings stabilizing, and similarly charged repulsive pairs destabilizing, coiled-coil conformation. Future mutational and/or structural work focused on the coiled-coil region could shed light on the nature of such critical interhelical interactions.

Our phylogeny informed approach to assign functions to certain BTN3A-regions allowed the identification of the 3A3_R381H mutant and a 3A1_3A3JM chimera as "super" BTN3A, merging the functions of heteromeric human BTN3A complexes in single, homomer-forming BTN3A molecules naturally occurring in the alpaca. The primordial BTN3A has been predicted to be a BTN3A3-like molecule with a functional

**Table 1 | ITC titration of BTN2A1 into PAg-bound BTN3A1**

| Titrant | Titrand | $K_D$ (µM) | $n$ | ΔH (kJ/mol) | ΔS (J/mol*K) |
|---|---|---|---|---|---|
| BTN2A1 ID271 | BTN3A1 BFI + HMBPP | 0.78 ± 0.28 | 0.94 ± 0.08 | −48.66 ± 2.11 | −45.62 ± 7.54 |
| BTN2A1 ID271 | BTN3A1 BFI + 4-M-HMBPP | 189.9 ± 174.8 | 0.08 ± 0.06 | −100 ± 0 | −260.4 ± 7.92 |

The binding parameters are obtained by independent fit using NanoAnalyze. Dates represent the mean ± SEM. (n = 3 independent experiments). BTN3A1 BFI: BTN3A1-full-length intracellular domain (JM+B30.2 domain); BTN2A1 ID271: BTN2A1-full-length intracellular domain (JM+B30.2 domain). "n" represents the number of molecules (titrant) interacts with titrand.

PAg-binding site that emerged with placental mammals[34,51,52]. This raises the question of what might have favored the evolution of BTN3A heteromers in primates[27] despite the efficacy of BTN3A homomers, as witnessed in alpaca[34]. Duplication of functional genes directly allows the acquisition of new features, even if these might have negative effects on the original function. This appears the case in humans, whereby the partnering BTN3A2 and BTN3A3 even lost PAg-binding function, which is compensated by the formation of new functional units via heteromerization with BTN3A1, thereby preserving the *BTN3A-TRGV9-TRDV* triad mandatory for PAg-sensing. One possibility is that devolving from a single BTN3A molecule, a substantial element of control of intracellular trafficking and IgV-related functionality may enable local fine-tuning of the strength of PAg-sensing via regulation of BTN3A2 and BTN3A3 expression. It will also be of interest to determine whether BTN3A1-JM might contribute to Vγ9Vδ2 T cell-independent features of BTN3A1, including ligation of CD45[53] or control of induction of type I interferon production by cytosolic TLR ligands[54].

Furthermore, it would be interesting to determine whether functional fusion proteins of different BTN relatives can also be achieved for the naturally occurring heteromers of Btn1/Btnl6, BTNL3/BTNL8, and Skint1/Skint2. Of note, such a fusion product is a frequently occurring copy number variation of *BTNL3* and *BTNL8*, resulting in the fusion of intracellular BTNL3 with the BTNL8 extracellular domain[55], which would be expected not to bind Vγ4-TCR[41]. This experiment of nature will allow testing of the physiological significance of the crosstalk, or the lack of it, between BTN(L) molecules and resolve the importance of TCR-BTNL3/8 binding for intestinal Vγ4 T-cell function and gut homeostasis and pathophysiology[56]. In addition, synthetic or natural "super" BTN3As, such as that of alpaca, might also be utilized as probes in the search for other factors involved in PAg-mediated Vγ9Vδ2 T cell activation.

A fourth key finding from our study was that we confirmed that HMBPP-binding to the BTN3A1 B30.2 domain promotes binding to the intracellular B30.2 domain of BTN2A1, and is consistent with our prior study[20] highlighting this interaction only occurs in the presence of a BTN3A1-B30.2-bound PAg such as HMBPP. Yuan et al. recently reported this interaction by size exclusion chromatography and an HMBPP coordinated complex consisting of an HMBPP-bound single BTN3A1-B30.2 domain and a dimer of BTN2A1 B30.2 domains[21]. Notably, our ITC data are consistent with that model because we observe a stoichiometry represented in terms of n value near 1, which may be expected if a dimer of BTN2A1 is interacting with a monomeric PAg-ligand-bound form of BTN3A1-B30.2. The importance of PAg-induced interaction between BTN3A1-ID and-BTN2A1-ID for PAg-induced activation is also in line with the observation that the HMBPP analog 4-M-HMBPP has a very poor stimulatory activity relative to HMBPP[46], as it does not support this interaction.

Based on these findings, we formulate the following working hypothesis as a model (Fig. 6). PAg-binding to the BTN3A1-B30.2 domain renders the BTN3A1-HMBPP complex into a ligand for the BTN2A1 intracellular domain. The function of the BTN2A1-V domain would be to recruit the TCR by binding to the CDR2 and HV4 regions of the TCRγ chain, and that of the BTN2A1 intracellular domain to recruit the HMBPP-bound BTN3A1-V. In the new complex, the binding of the TCRγ (CDR2 and HV4) chain to the C-F-G surface of the BTN2A1-V domain would be retained, while other CDRs might additionally interact with the newly formed BTN2A1-BTN3A complex as proposed

recently[57]. A direct interaction of the Vγ9Vδ2 TCR with V-domains of BTN2A1-BTN3A complexes would also be compatible with a most recent report that shows direct stimulation of Vγ9Vδ2 T cells by recombinant BTN3A1-BTN2A1 heteromers in the presence of a co-stimulus[58]. However, it is yet to be proven whether BTN2A1 and BTN3A1 can form a functional heterodimer. In conclusion, our results support a composite ligand model we first proposed following the identification of BTN2A1 as a ligand for germline-encoded regions of the Vγ9 TCR chain[13]. They are also entirely consistent with our recent study indicating such a mechanism could allow inside-out signaling initiated by PAg-induced association of the intracellular domains of BTN3A and BTN2A1 molecules[57]. Both Willcox et al.[57] and the current study support the idea that by triggering the formation of a composite TCR ligand comprising BTN2A1 IgV juxtaposed to BTN3A1 IgV domains (or alternatively BTN2A1 IgV and BTN3A1 IgV plus hypothetical ligand), intracellular PAg binding would ultimately coordinate extracellular engagement of both germline-encoded regions and additional CDRs of the TCR. Our current results extend this model substantially by clarifying the critical role heteromeric partners (BTN3A2 and BTN3A3) of BTN3A1 play in the PAg sensing process, in facilitating interactions that are topologically optimized to surpass the requirements to initiate TCR signaling (Fig. 6). The specific role of the BTN3A2 or BTN3A3 molecules in this complex would be in forming BTN3A1-containing heteromers that adopt a unique surface topology most probably different from BTN3A1-homomers. Second, such heteromers still allow PAg-binding to the B30.2 domain of BTN3A1 and its subsequent interaction with B30.2 domains of the BTN2A1 homodimer[21,59], resulting in a complex with a distinct topology of the BTN3A-and BTN2A1-V domains supporting a stimulatory TCR-engagement.

Interestingly, BTN3A1-induced immunosuppression was reported for BTN3A1 overexpressing tumor cells, and it is abolished by BTN3A1-V domain-specific mAbs and by Zoledronate[53] It will be interesting to learn whether V-domain interactions between constitutively expressed BTN2A1 and BTN3A[13,57] or newly formed PAg-induced BTN3A-BTN2A1 complexes[13,20,35,59] might affect such suppression.

The scenario discussed above is hypothetical and final clarification of the exact nature of the ligand recognized by the Vγ9Vδ2 TCR during PAg-activation has still to be elucidated. Nevertheless, the data we present and the molecular ground rules they formulate will be instrumental in guiding future studies to resolve this problem.

## Methods
### Experimental models and cell lines
53/4 hybridoma TCR transductants were cultured with RPMI (Gibco) supplemented with heat-inactivated 10% FCS, 1 mM sodium pyruvate, 2.05 mM glutamine, 0.1 mM nonessential amino acids, 5 mM β-mercaptoethanol, penicillin (100 U/mL) and streptomycin (100 U/mL). Peripheral blood mononuclear cells were isolated from healthy volunteers. They were also maintained with the abovementioned medium with or without rhIL-2 (Novartis Pharma). 293T cells were maintained in DMEM (Gibco) supplemented with 10% FCS.

For further information and requests for reagents, please contact the lead author.

### Generation of 293T BTN3AKO cells
293T BTN3KO (3KO) and BTN3A1KO (A1KO) cells used were mentioned in our previous study. The BTN3A2KO (A2KO), BTN3A3KO

γδ T cells

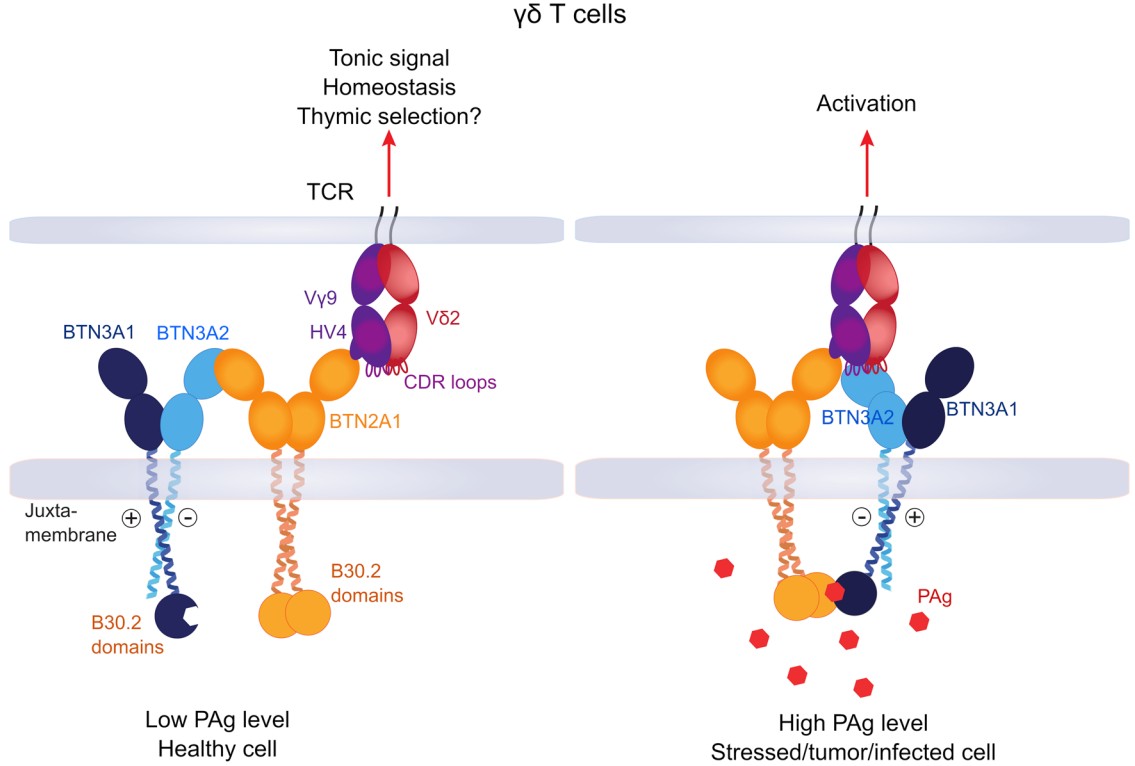

**Fig. 6 | PAg induced Vγ9Vδ2 T cell activation by BTN3A-BTN2A1 composite ligand.** In a resting state of the target cell, the heteromeric BTN3A (BTN3A1-BTN3A2/BTN3A3) interacts with BTN2A1 via their V-domains, and the BTN2A1-V domain interacts with germline-encoded HV4 and CDR2 regions of Vγ9 chain of Vγ9Vδ2 TCR. Such interaction may act like a tonic TCR signal for maintaining homeostasis or even could be involved in the thymic selection of T cells. However, in case of stress in the target cell, the accumulated PAg (red) binds to the B30.2 domain of BTN3A1, which further interacts with the B30.2 domains of BTN2A1. Consequently, the heteromeric JM region in the BTN3A complex permits the formation of appropriate topology of the V-domain of partnering BTN3A (BTN3A2/BTN3A3) distal to the PAg-B30.2 domain of BTN3A1. This complex, on its own or in combination with an unknown hypothetical ligand, combines molecular interactions mediated by both BTN2A1 and BTN3A2/A3 with the TCR, surpassing the threshold for TCR triggering to permit γδ T cell activation.

(A3KO) and BTN3A2 & BTN3A3KO (A2A3KO) cells were also generated as previously reported[34]. The CRISPR sequences and the primers used for the validation of KO with genomic DNA are mentioned in Supplementary Table 1.

### Generation of BTN3A, tagged BTN3A and BTN3A-fluorescent protein constructs

The full-length BTN3A1 and BTN3A1-mCherry fusion construct were generated as mentioned previously[34]. The full-length BTN3A2 and BTN3A3 were subcloned from previously reported pIRES1hyg vectors[15]. For the generation of pIH-FLAG, pIH vector[60] was digested with EcoRI and BamHI. Sequentially, the insert with MfeI and BglII restriction sequences as 5′ and 3′ overhangs that comprise BTN3A1 leader sequence followed by FLAG sequence, linker sequence, and restriction sites for BamHI and EcoRI was digested with MfeI and BglII and cloned to EcoRI-BamHI digested pIH vector. This vector was further digested with BamHI and EcoRI and used to clone the desired BTN3A sequence from IgV to stop codon or IgC to stop (VΔ3A1 or VΔ3A2) sequence. pIZ-HA-tagged BTN3A1 or BTN3A1_A3JM was generated with EcoRI and BamHI digested pIZ vector[60]. Two PCR products with overlapping overhang sequences in which product 1, BTN3A1 leader sequence followed by HA tag and linker sequence (used above) and product 2, BTN3A1-IgV-domain till stop codon were cloned into the above-digested pIZ vector using In-Fusion HD cloning (TAKARA) as per manufacturer's instruction. The BTN3A1_A3JM chimera was subcloned from below below-mentioned pCDNA 3.1 vector. The multiple cloning site sequences pIH-FLAG and pZ-HA are provided in Supplementary Table S1. GeneArt gene synthesis (ThermoFisher Scientific) synthesized the full-length BTN3A_JM chimeras by swapping the nucleic acids encoding for the JM region (272–340 amino acid[36]) between BTN3A1 and BTN3A3. The JM chimeras cloned in the pCDNA 3.1 vector were provided by the manufacturer, and JM chimeras were further subcloned into the phNGFR linker mCherry vector. phNGFR linker mCherry was used as the backbone to generate phNGFR linker CFP and phNGFR linker YFP, to which FLAG-3A1 or FLAG-VΔ3A1 and BTN3A1 or BTN3A1_A3JM chimera was subcloned, respectively. NEB 5-alpha (NEB) was used as a transformant of the abovementioned plasmids. The plasmids cloned with wild-type BTN3A proteins or mutant BTN3A were expressed in 293T 3KO via retroviral transduction[61]. All the restriction enzymes were purchased from Thermo Fisher Scientific. All the plasmids and cloned corresponding constructs were mentioned in Supplementary Table 2.

### In vitro stimulation of human Vγ9Vδ2 TCR transductants

In this, $1 \times 10^4$ 293T (DSMZ, ACC 635) or KO and their BTN3A transductants were seeded in 50 μL DMEM medium in 96 well flat-bottom tissue culture plate on day 1 and incubated overnight. On day 2, 50 μL of 53/4 r/mCD28 human Vγ9Vδ2 TCR transductants (MOP)[24] at $1 \times 10^6$ cells/mL density and 100 μL of HMBPP (SIGMA, 95058) at mentioned concentrations were added to the culture and incubated for 22 h at 37 °C. Post 22 h, the activation of TCR reporter cells was measured by analyzing the supernatants of cocultures for mouse IL-2 via ELISA (Invitrogen, 88-7024-88) as per the manufacturer's protocol.

## Expansion of primary polyclonal human Vγ9Vδ2 T cells

Fresh peripheral blood mononuclear cells (PBMCs) were obtained from healthy volunteers with informed consent, according to the University of Wuerzburg institutional review board (Gz. 20220927 01). Tubes preloaded with Histopaque-1077 (SIGMA, 10711) were layered with whole blood and centrifuged at 400×g for 20 min at room temperature with no acceleration or brakes. The opaque interface containing PBMCs was aspirated after centrifugation and was washed twice at 461×g for 5 min. PBMCs were cultivated with RPMI containing heat-inactivated 10% FCS, 100 IU/mL recombinant human IL-2 (Novartis Pharma) and 10 nM HMBPP in $10^6$ cells/mL density in a 96-well plate round bottom plate. After 10 days, cells were pooled and washed twice and cultured in a 6-well plate in $10^6$ cells/mL for 3 days without rhIL-2. Such rested cells were subjected to further experiments.

## Human polyclonal Vγ9Vδ2 T cell activation assay

293T cells at $2 \times 10^4$ cells/100 µL (DMEM, 10% FCS) per well were cultured in triplicates in 96-well plate flat bottom with or without 25 µM zoledronate (SIGMA) overnight. The next day, cells were washed twice with PBS, and Vγ9Vδ2 T cells expanded from PBMCs at $2 \times 10^4$ cells/100 µL per well were added and cultured for 4 h. After 4 h, supernatants were frozen at −20 °C until human INFγ assay ELISA (Invitrogen, EHIFNG) could be performed as per the manufacturer's instructions. For the CD107a assay, 293T cells were seeded as abovementioned. Vγ9Vδ2 T cells expanded from PBMCs were also added as abovementioned but along with anti-CD107a-PE (1:200; BD Pharmingen) conjugated antibody and cultured for 4 h. After 4 h, the cells were collected from the wells as triplicates and washed once with PBS. After which cells were treated with anti-human Vδ2-FITC (1:200; Beckman Coulter) conjugated antibody for 20 min and washed once, followed by analysis at FACSCalibur (BD) for the percentage of Vδ2-FITC and CD107a-PE population.

## Flow cytometry for surface and total expression of BTN3As

293T and 3KO transductants of BTN3As (WT and Chimaeras) were acquired by FACScalibur (BD) and analyzed with FlowJo. For total staining, cells were fixed with fixation buffer for 30 min at RT, followed by wash and incubation for 30 min with permeabilization buffer at RT. Then cells were stained with antibodies that were prediluted in permeabilization for 30 min at 4 °C, as per the manufacturer's instructions (eBiosciences, eBiosciences™ Intracellular Fixation & Permeabilization buffer set). For surface staining, cells were directly stained with antibodies of interest for 30 min at 4 °C. The BTN3As were detected by unconjugated mAb 103.2 (1 µg/mL; gift from Daniel Olive). If tagged, unconjugated anti-FLAG (1:1000; M2, SIGMA) and anti-HA (1:250; F-7, Santa Cruz) antibodies were used. The primary antibodies were detected by F(ab')2 Donkey anti-mouse IgG (H+L)-APC (1:500; Jackson Immunoresearch, 115-136-146). mIgG1k and mIgG2a k (1:500; eBiosciences) were used as isotype controls.

## Immunoprecipitation

In this, $3 \times 10^6$ cells of 3KO and BTN3A-transductants were seeded in a 10 cm tissue culture plate on day 1. On day 3, the cells were lysed with 400 µL of lysis buffer[33] [(50 mM Tris·HCl at pH 7.4, 150 mM KCl, 10 mM $MgCl_2$, 1 mM $CaCl_2$, 0.5% Nonidet P-40, 0.1% digitonin, 5% glycerol, Complete Protease inhibitor(Roche)]. The lysate was rigorously vortexed for 15 min at 4 °C and was centrifuged at 21,000 x g rpm for 15 min at 4 °C. After centrifugation, 50 µL lysate was kept aside as input. The remaining lysate was incubated for 4 h at 4 °C with 50 µL of protein-G Sepharose™ (GE, 1706180) beads complexed with anti-FLAG (1:1000; M2 clone, SIGMA) and washed thrice with lysis buffer. Proteins were eluted with 80 µL of Laemmli and analyzed by SDS-PAGE and Western Blotting. The blots were treated with anti-Vinculin (1:2000; SIGMA), anti-FLAG and anti-HA (1:2000; CST) as primary antibodies

overnight at 4 °C. The following day, the blots were washed thrice and treated with protein-A-HRP (1:2500; SIGMA) conjugate for an hour at RT and washed and developed with Pierce SuperSignal™ West Femto Maximum Sensitivity Substrate (Thermo Fisher Scientific). The blots were visualized with the LI-COR Odyssey imaging system.

## Blue native gel electrophoresis

Blue native gel electrophoresis was performed as described in ref. 62.

## Immunofluorescent staining

293T, 3KO and 3KO-BTN3A transductants were seeded in $5 \times 10^4$/200 µL in Ibidi 8 well µSlides on day 1. On day 2, for live-cell imaging, cells were washed twice with PBS and treated with anti-FLAG (1:1000; M2 SIGMA) or anti-HA (1:1000; CST) for 20 min, followed by three washes and treated with anti-mouse AF647 (1:500; Invitrogen) or anti-Rabbit AF555 (1:500; Invitrogen) for 30 min. After 30 min, cells were washed thrice and visualized with confocal microscope Zeiss LSM 780 under 63x (NA 1.4) oil immersion lens with 514 and 633 lasers. Acquired images were further analyzed using ImageJ. For fixed cell imaging, the cells were fixed with 4% paraformaldehyde for 30 min and either treated with 0.1% TritonX-100 for permeabilization or treated with anti-FLAG or anti-HA antibodies overnight. The following day, cells were washed and treated with anti-mouse AF648 or anti-rabbit AF565 for 1 h and washed thrice, followed by incubation with BODIPY-FL_DHPE (1:500; Invitrogen) for 10 min on ice and 10 min at RT and washed thrice before acquiring images under the microscope as above.

## Fluorescence recovery after photobleaching

293T and 3KO transduced with 3A1 mCherry fusion construct (3A1-mC) or 3A3_R381H-mC or their JM chimeric constructs were seeded in Ibidi 8well µSlides at $5 \times 10^4$/200 µL per well on day 1. On day 2, cells were analyzed with a confocal microscope Zeiss LSM 780 under a 63x (NA 1.4) oil immersion lens with a 560 laser. The rectangular regions were marked on the cells of interest, the marked regions were photobleached with 100% laser energy for 5 s (>90% loss of fluorescence). Images were collected every 5 s after photobleaching for 100 s. The percentage of the immobile fraction was derived from the belowmentioned formula

Mobile fraction $F_m = (I_E − I_0) / (I_I − I_0)$; Immobile fraction $F_i = 1 − Fm$; where: $I_E$: End value of the recovered fluorescence intensity, $I_0$: first postbleach fluorescence intensity, $I_I$: Initial (prebleach) fluorescence intensity.

## Fluorescence resonance energy transfer

3KO transduced with FLAG-BTN3A1-CFP or FLAG-VΔ3A1-CFP and BTN3A1-YFP or BTN3A1_A3JM YFP constructs were plated over the glass coverslips. Before imaging, cells were incubated in the imaging medium (144 mM NaCl, 5.4 mM KCl, 1 mM MgCl2, 1 mM CaCl2, 10 mM HEPES; pH = 7.4) and mounted on a Leica DMI 3000 B microscope fitted with a 63x/1.40 objective. The cells were excited with CoolLED (440 nm), and the emission light was split into donor and acceptor channels using the DV2 QuadView (Photometrics) equipped with the 505dcxr dichroic mirror and D480/30 m and D535/40 m emission filters. When CFP and YFP are in FRETable distance, the emitted light detected by 535 filters (YFP) would be greater than 480 filters which can be presented as pseudo-colored ratio images with a reference FRET ratio (FR) chart. Images were acquired using a CMOS camera (OptiMOS, QImaging) and Micro-Manager 1.4. software was used for data analysis[63,64].

## Synthesis of 4-M-HMBPP

The binding of 4-hydroxy-3-(4-methylbenzyl)but-2-en-1-yl diphosphate (4-M-HMBPP) to BTN3A1 was previously described by Yang et al.[46], but the synthetic route has not yet been reported. We adapted the method of Yang et al. (Yonghui Zhang, personal communication to T.H.) to

obtain 4-M-HMBPP as detailed in the supplementary text for use in these studies.

## Isothermal titration calorimetry (ITC)

ITC was performed as described[20] using a nanoITC (TA Instruments). The concentrations of the titrant and titrand are indicated in the figure legend.

## Modeling BTN3 juxtamembrane coiled-coil dimers

Models of the juxtamembrane (JM) coiled-coil dimers were generated using the CCBuilder2 server (http://coiledcoils.chm.bris.ac.uk/ccbuilder2/builder)[45]. Models were generated using default settings assuming a parallel homo/hetero dimeric structure, encompassing residues Q273–L312 for human BTN3A1, BTN3A2, and BTN3A3 and alpaca BTN3A3. BTN3A1 was modeled with Q273 at the "c" position of the heptad repeat, whereas all other BTN3 molecules were modeled with Q273 at the "d" position. Models of human BTN3 proteins were further refined using the "Optimize" function of the CCBuilder2 program. JM coiled-coil dimer interface contacts were determined using the program NCONT as part of the CCP4 suite[65]. Structural figures were generated using PyMol[66].

## Statistics

Statistical analysis of stimulations was performed with GraphPad Prism using ordinary two-way ANOVA, and all stimulation data sets were representatives of three independent experiments (n-3), their mean was shown in the form of column graphs or XY graphs with error bars representing the standard deviation (SD). Similarly, samples analyzed for FRAP were subjected to multiple $t$-tests and statistical significance was determined using the Bonferroni-Dunn method, and SD was shown as error bars. The normalized FRET data was analyzed with ordinary one-way ANOVA and SD was shown as error bars.

## Data availability

All the data supporting the findings of this study are available within the article and its supplementary information files. The original data of all the experiments are provided in the source data in this paper.

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

## Acknowledgements

We would like to thank Christian Hackenbroch for the generation of BTN3A_JM chimera and JM mutants. We would like to thank the Core Unit for FACS of the IZKF Würzburg for supporting this study. We also thank Christine Krempl from the Institute for Virology and Immunobiology for maintaining the confocal microscope. This work was supported by grants to A.J.W. by the United States National Institutes of Health (AI150869 and CA266138); T.H. by Wilhelm–Sanderstiftung, Germany (grant 2013.907.2), German Research Foundation (DFG, HE2346/8-2 within FOR 2799 "Receiving and Translating Signals" via the γδ T Cell Receptor); B.E.W. by the Wellcome Trust, United Kingdom (Investigator Award 221725/Z/20/Z supporting C.R.W. and F.M.); to W.W.S. by the DFG through BIOSS - EXC294 and CIBSS - EXC 2189, SFB1381 (A9) and SCHA-976/8-1 within FOR 2799 "Receiving and Translating Signals" via the γδ T Cell Receptor; to C.J. by the DFG through GSC-4 (Spemann Graduate School) and to V.O.N. by the DFG SFB1328 "Adenine nucleotide in immunity and inflammation".

## Author contributions

Conception of the study: T.H., M.M.K. Design of experiments: M.M.K., H.S., F.M., B.K., C.J., R.S., W.W.S., A.J.W., B.E.W., T.H. Acquisition of data: M.M.K., H.S., F.M., Y.J., B.K., C.J., L.S., N.L., R.S. Analysis of data: M.M.K., H.S., Y.J., F.M., B.K., L.S., R.S., W.W.S. Interpretation of data: M.M.K., H.S., Y.J., F.M., B.K., C.J., C.R.W., R.S., V.O.N., V.K., A.J.W., B.E.W., T.H. First draft of the manuscript: M.M.K., T.H., B.E.W. Substantial revisions: M.M.K., B.E.W., T.H.

## Funding

## Competing interests

B.E.W. provides consultancy regarding the development of γδ T cell immunotherapy approaches for Ferring Ventures SA, linked to Ferring Pharmaceuticals. T.H. is supported by Byondis B.V. for work not related to this study.

## Additional information

[1]Institute for Virology and Immunobiology, University of Würzburg, Würzburg, Germany. [2]Institute of Experimental Cardiovascular Research, University Medical Center Hamburg-Eppendorf, Hamburg, Germany. [3]DZHK (German Centre for Cardiovascular Research), Partner Site Hamburg/Kiel/Lübeck, Hamburg, Germany. [4]Institute for Systems Genomics, University of Connecticut, Storrs, CT 06269, USA. [5]Cancer Immunology and Immunotherapy Centre, Institute of Immunology and Immunotherapy, University of Birmingham, Edgbaston, Birmingham, UK. [6]University Hospital Wuerzburg, Department of Internal Medicine II and Comprehensive Cancer Center (CCC) Mainfranken Wuerzburg, Wuerzburg, Germany. [7]Signaling Research Centers BIOSS and CIBSS, University of Freiburg, Freiburg, Germany. [8]Department of Immunology, Faculty of Biology, University of Freiburg, Freiburg, Germany. [9]Centre for Chronic Immunodeficiency (CCI), Faculty of Medicine, University of Freiburg, Freiburg, Germany. [10]Spemann Graduate School of Biology and Medicine (SGBM), University of Freiburg, Freiburg, Germany. [11]Department of Pharmaceutical Sciences, University of Connecticut, Storrs, CT 06269, USA. [12]Department of Pharmaceutical Sciences, School of Health Sciences & Technology, Dr. Vishwanath Karad, MIT World peace University, Pune 411038, India.
✉e-mail: mohindar.karunakaran@uni-wuerzburg.de; thomas.herrmann@uni-wuerzburg.de

