## [Peer Review File · Nature Communications]

REVIEWER COMMENTS

Reviewer #1 (expert in $\gamma\delta$ T cell activation):

This is an extensive and comprehensive molecular analysis of the differential roles of structural elements (V, JM, B30.2 domains) of BTN3A species and BTN2A1 as to how they act in concert to allow activation of human Vg9Vd2 T cells by phosphoantigens. As such, this study adds new and important information. The presented results support their model depicted in Fig. 6, in which pAg binding to B30.2 of BTN3A1 which then interacts with B30.2 of BTN2A1 with subsequent critical involvement of the BTN3A JM region and V domains of BTN3A2/A3. While it still remains unclear at this point how the activation of the T-cell receptor is brought about (there is still room for a so far unidentified "ligand" which could play a role). Overall, the authors are to be congratulated to this work, the conclusions (and the model) are based on sophisticated molecular analyses, imaging and modeling.

Specific comment1:

1. The basic (and very "clean") system was to use 293T cells as recipients of all kind of variant BTN3A/2A constructs, and then using HMBPP to activate BTN and subsequently Vg9Vd2 TCR, either TCR transfectants of expanded human Vg9Vd2 T cells. Significant results were obtained in all instances with very high concentrations of HMBPP (10 and 1 micromolar), whereas lower concentrations (0.1 micromolar) – which are perfectly able to activate Vg9Vd2 T cells among e.g. PBMC (this also works with 10 and 1 nM!) - showed very little if any activity (e.g. Fig. 1a, 1c, 1f, 1h; Fig. 3a, 3f; Fig. 4c). Therefore, the authors should discuss how the results obtained in their system requiring such (very) high concentrations of pAg can translate into relevance for the physiological situation where much lower concentrations of pAg are potent $\gamma\delta$ T-cell activators.
2. They should also comment on how HMBPP added exogenously to the cell culture can activate the intracellular B30.2 domain. How does HMBPP get into the cell? Is the entrance into 293T cells equally efficient as into "normal" cells (e.g. monocytes)?

Minor:

1. Line 146: should read "stronger" rather than "strong"?
2. Line 579: Here they used BrHPP, while in all other assays they used HMBPP. Please explain abbreviation BrHPP and state the source.
3. Line 987: CRR should read CDR
4. Lines 682-690 (references): I would suggest to include more recent papers, e.g. DOI: 10.1038/s41423-020-0504-x DOI: 10.1038/s41571-022-00722-1

Reviewer #2 (expert in immunogenetics of antigen presentation):

This manuscript explores aspects of the interaction between the juxta-membrane (JM) regions of butyrophilin (BTN) 3A1, 3A2 and 3A3 in homo- and heterodimers as assessed by cell surface expression, stimulation of an appropriate gamma-delta T cell reporter line and by selective mutation and structural modelling, along with the requirements for the location of V and B30.2 domains (Figs. 1-4). In addition, the authors examine the effect of a non-stimulatory phosphoantigen (pAg) on the interaction of the intracellular regions of 3A1 and 2A1 (Fig. 5). On this basis, the authors propose an overarching model to explain many disparate findings in the literature and in their own manuscript (Fig. 6).

To a reviewer familiar with the general outlines of BTN research over the last years, but without detailed knowledge of some recent findings (including the latest papers, preprints, manuscripts in preparation and personal communications cited in the text), the present manuscript seems to add important new experiments and interpretations to the growing literature about BTN-family function and mechanism, and contributes a model incorporating as much information as possible. However, there are some concerns to be addressed.

In a general sense, the results in Figs. 1-3 are detailed and convincing. The weakest part of the manuscript (in the opinion of this reviewer) is Fig. 4, in which the authors pick out one sequence

feature from the JM region, then test sequence swaps (which are good experimental data) and finish with a series of structural predictions which are reasonable but only speculation without further experimental verification. Moreover, the authors do not explain why they chose this particular sequence feature and indeed point out that other parts of the JM regions must be important, without testing other sequence portions. Fig. 5 is convincing work, but all but one panel is already reported in the paper of one of the co-authors (Hsaio et al 2022 Cell Chem Biol), with the new data in this manuscript examining the effect of 4-M-HMBPP, which is an important finding (T cell stimulation correlates not with PAg binding but with binding followed by B30.2 domain interaction between 3A1 and 2A1). (Parenthetically, the authors seemed to do a good job of crediting other labs for published and unpublished results leading up to the experiments in this manuscript.) Finally, Fig. 6 is an excellent model, but there may be alternatives. For example, the model proposes that upon binding of PAg to the B30.2 domain of BTN3A1-3A2/3 heterodimer, their interaction of the V region of BTN2A1 with the TCR (which is not enough to stimulate Vgamma9Vdelta2 T cells) is supplemented by binding of BTN3A2/3 to the TCR which is enough for stimulation. The authors cite one short study from a company as well as unpublished data by others as evidence for T cell stimulation, but otherwise do not explain why they prefer this model to the idea that BTN2A1 binding to TCR is inhibitory for the T cell, and that PAg binding to BTN3A1 leads to BTN2A1 to stop binding the TCR, releasing the inhibition and allowing the TCR to bind an appropriate stress ligand on the target/antigen-presenting cell. Perhaps the answer is well-known to members of this scientific community, but it seems unclear at least to this reviewer.

The fact that the authors have written up and submitted their results is understandable, given the number of preprints, manuscripts in preparation and personal communications cited in the text, and of course they wish to get their story out before someone else says more-or-less the same things. However, is there a reason not to utilise the preprint server bioRxiv which would give their manuscript instant priority and (presumably) protect their submission to Nature Comms, while allowing them to finish the detailed experiments which would strengthen their conclusions? Of course, such assurances are between the communicating author and the editor, and are beyond the understanding of this reviewer.

In addition, there are detailed concerns below.

Line 55, 222. "heteromers" and "homomers", "homomers or heteromers". These terms are used along with homodimers and heterodimers. Are these terms being used interchangeably, or is there a deeper meaning obvious to members of this scientific community (for example, homomers may include dimers, trimers and other higher order interactions). The simplest clearest usage is to be preferred, if possible.

Lines 200-204, Figure 2b. "Under fixed-permeabilized conditions (right hand panels) FLAG-Vdelta3A1 was detectable at the cell surface only if colocalizing with HA-3A1_A3JM (violet, right). In contrast to live conditions, clear colocalization of FLAG- FLAG-Vdelta3A1 and HA-3A1 was observed in cytoplasmic vesicles. Notably, when FLAG-Vdelta3A1 was coexpressed with HA-3A1_A3JM, HA-tag was detected largely at the membrane, with hardly any detectable in cytoplasmic vesicles." This reviewer saw lots of violet staining apparently at the cell surface in the center panel headed with FLAG-Vdelta3A1 plus HA-3A1, but struggled to see the very little violet staining in the far-right panel with FLAG-Vdelta3A1 plus HA-3A1_A3JM, which had lots of apparently intracellular vesicles with separate red and blue staining. These confocal images seem to be at odds with the flow cytometry in Fig. 2a, where only the FLAG-Vdelta3A1 plus HA-3A1_A3JM was well-expressed at the cell surface. Please explain. Were the panels switched, or is this reviewer simply misunderstanding the data?

Lines 322-237, Figure 2f. "However, 3A1-CFP co-expressed with 3A1_A3JM-YFP revealed high FRET predominantly at the membrane (Fig. 2f, upper right) ... Even stronger FRET was observed at the membrane when FLAG-Vdelta3A1-CFP was co-expressed with 3A1_A3JM-YFP (Fig. 2f, lower right) ...". In the two upper panels, the interactions are depicted as green (intermediate interaction according to the scale), but in the two lower panels, one is green and the other red (the highest level interaction). There is no indication of how representative these four panels (with two cells each) might be, and perhaps the authors might consider some form of quantitation from many such panels.

Lines 244-245, Figure 2e. "... suggests the formation of heteromers in which their respective B30.2 domains are in FRET-able distance as predicted (Fig 2e)." The left-hand side of Fig. 2e with different models of V domain association is not discussed, and the right-hand side with different models of TM-JM-B30.2 association with regard to the ectodomains is not discussed, in particular to what has been learned from the FRET experiments. Did the authors wish to say that their findings of FRET-positive interactions are consistent with two models: the V-C interactions of ectodomains which lead to TM and JM far apart but B30.2 domains close, or with the C-C interaction of ectodomains which lead TM-JM-B30.2 all being close? Or what is intended here?

Lines 267-269, Figure 3d. "... we tested the effects ... on the cell surface immobility of 3A-molecules by FRAP (Fluorescence Recovery after Photobleaching)." Some introduction about what the meaning of immobility under these conditions would be very helpful for those who are not aware of what the point of the experiment might be.

Line 281. "... two complexes of less than 44 kDa apparent MW." Molecular mass (Mr) is now considered the appropriate term (MW referring to weight which depends on gravity, not an intrinsic property of the molecule).

Lines 292-296, Figure 3e. "To test whether similar effects were also observed for a homomeric ..., a mutant ... was generated co-expressed (Fig. 3e)." Despite looking and thinking, the point of showing Fig. 3e was unclear to this reviewer. Please explain a bit more in the text.

Lines 296-298, Figure 3g. "Stimulation was successfully detected with both the co-transductants where the PAg-binding site and wild type V-domain were located on different molecules (Fig. 3f-g) ...". Some aspects of this panel are not clear to this reviewer, and are not described in the text or figure legend. What is the difference between the dark blue and light blue V domains, the three colours of JM, and the arrow pointing up versus the "T" at the end of the brown line. Also, why is only one PAg shown bound to each of the B30.2 dark orange domains; is this meant to imply that PAg can only bind to one of the two B30.2 domains? Finally, what is this panel supposed to represent, predictions or experimental results? If it is supposed to represent the true situation from experiments (implied from "observed outcomes" in the figure legend), where is the data?

Lines 305-308 and 313-314, Figure 4a. "We noted that the JM of BTN3A1 contains a positively charged lysine triplet (KKK) ... two negatively charged glutamic acid residues (ExE) at this position (Fig. 4a) ... suggesting other regions of the JM may also be involved in controlling cooperation and trafficking of associated 3A-molecules ...". There is no indication of why the authors focused on such a small feature of the sequences which differ in so many other places (and places in which the authors themselves describe as having effects). Only one feature was examined functionally, and all the rest is predicted models, without any supporting experimental data, but with a huge amount of speculation. Moreover, there is no consideration of what conformational changes might occur when PAg binds to the B30.2 domain or when the 3A1 B30.2 domain binds the 2A1 B30.2 domain. A full set of experiments swapping sections or scanning by alanine/serine substitution would seem to be warranted, or at the least functional and structural predictions from swapping the other sections mentioned. Perhaps the authors can justify these experiments as the starting-point for a full investigation, if the rationale for picking this little sequence feature can be explained.

Lines 284-285, Figure 5. "Here, we examined the ability of 4-M-HMBPP to support this interaction." The text was totally opaque to this reviewer (including the description of "titrated BTN2A1 intracellular domain") until the figure was examined. Perhaps the experiment would be clearer if the sentence was modified to say "Here, we examined the ability of 4-M-HMBPP to support this interaction by expressing and purifying the full intracellular domains of BTN3A1 (construct BFI) and BTN2A1 (construct ID271), and then using isothermal calorimetry (ITC) to measure association." Also, how was "Titration of 960 microM 4-M-HMBPP into the buffer" (panel a) carried out, what is meant by "Mole Ratio" on the x-axis of panel a, and how was the concentration of 4-M-HMBPP for the rest of the panels decided? Also, rather than simply a, b, c, d and e above the panels in the figures, the authors could allow the readers to understand much more easily if some descriptive title for each panel was shown (much like the other figures).

Lines 399-400. "Firstly, it demonstrates a crucial role for the V-domain for cell surface expression of BTN3A molecules." It seemed unclear to this reviewer whether the authors meant a particular 3A chain, or the heterodimers/heteromers. Perhaps the sentence might be "Firstly, it demonstrates a crucial role for the V-domain for cell surface expression of BTN3A chains." or "Firstly, it demonstrates a crucial role for at least one V-domain for cell surface expression of BTN3A molecules."

Lines 497-502, Figure 6. "The function of the BTN2A1-V domain would be to recruit the TCR by binding to the CDR2 and HV4 regions of the TCRgamma chain ... while other CDRs might additionally interact with the newly formed BTN2A1-BTN3a complex that is in line with the finding of the Willcox research group (Willcox, C. R. et al 2023, in preparation)." The authors do not mention an alternative model in which BTN binding to the TCR is inhibitory; do they believe that this model of inhibition is no longer viable? If the data in this manuscript contributes to the choice between models, should this be considered in the Discussion?

Reviewer #3 (expert in butyrophilins):

This article finetunes some aspects of the cooperation between different butyrophilin proteins. Overall for the community not so much news and in my perception more for a specialized journal and would benefit from some more stringent writing. Also the translational relevance of the findings is missing.

Some minor points picked up

- Fig. 1F: symbols not well visible
- Fig. 3D: Concentration of PAM and BT3 20.1 not mentioned (neither in figure, text or methods)
- Fig. 5: To me, this is the weakest part of the article.
 - o I would suggest to put headings to the graphs for a better overview. I think the line in (5a) is not correctly placed.
 - o Might be just me and the fact that I'm not used to interpreting this kind of experiments and graphs, but this part and figure took me quite long to really get through and understand why they can make certain statements. That BTN2A1 ID271 and BTN3A1 BFI are only interacting under pAg, has been shown in their reference already, and they didn't state any physiological relevance why to investigate 4-M-HMBPP instead of HMBPP. Is it an important competitor or very abundant?
 - o In their discussion they say "Notably, our ITC data are consistent with that model because we observe an n value near 1,.." (p. 18), but don't explain what the n value is and what conclusions you can draw from this.
 - o To me this part also comes a bit out of the blue and I don't see a clear link of this 4-M-HMBPP to the rest of the story.. Relevance of 4-M-HMBPP physiological conditions is also not being stated later on.
- Table 1 is missing a good table legend and in my opinion a more elaborate explanation.
- Please comment: Is there a reason why the experiments besides Stimulation assays have been performed in the absence of ABP?

We thank all reviewers for their swift, careful and helpful comments. We also like to thank the editor for providing an opportunity to revise the manuscript. We have provided point-by-point responses below in ARIAL font. The changes are also tracked in the manuscript.

RESPONSE TO REVIEWERS' COMMENTS

Reviewer #1 (expert in $\gamma\delta$ T cell activation):

This is an extensive and comprehensive molecular analysis of the differential roles of structural elements (V, JM, B30.2 domains) of BTN3A species and BTN2A1 as to how they act in concert to allow activation of human Vg9Vd2 T cells by phosphoantigens. As such, this study adds new and important information. The presented results support their model depicted in Fig. 6, in which pAg binding to B30.2 of BTN3A1 which then interacts with B30.2 of BTN2A1 with subsequent critical involvement of the BTN3A JM region and V domains of BTN3A2/A3. While it still remains unclear at this point how the activation of the T-cell receptor is brought about (there is still room for a so far unidentified “ligand” which could play a role). Overall, the authors are to be congratulated to this work, the conclusions (and the model) are based on sophisticated molecular analyses, imaging and modeling.

We thank the reviewer very much for his/her positive and encouraging comments. The two specific questions raised touch on an important issue and as outlined below we have now included related references and short explanatory sentences in the manuscript to address them. A more detailed and specific answer for the reviewer is given below.

Specific comment1:

1. The basic (and very “clean”) system was to use 293T cells as recipients of all kind of variant BTN3A/2A constructs, and then using HMBPP to activate BTN and subsequently Vg9Vd2 TCR, either TCR transfectants of expanded human Vg9Vd2 T cells. Significant results were obtained in all instances with very high concentrations of HMBPP (10 and 1 micromolar), whereas lower concentrations (0.1 micromolar) – which are perfectly able to activate Vg9Vd2 T cells among e.g. PBMC (this also works with 10 and 1 nM!) - showed very little if any activity (e.g. Fig. 1a, 1c, 1f, 1h; Fig. 3a, 3f; Fig. 4c). Therefore, the authors should discuss how the results obtained in their system requiring such (very) high concentrations of pAg can translate into relevance for the physiological situation where much lower concentrations of pAg are potent $\gamma\delta$ T-cell activators.

Our study analyzes cellular and molecular events underlying TCR-mediated activation by pAg such as HMBPP and the role of BTNs in this process. This restricts the choice of responders and stimulatory cells as outlined below. We acknowledge that our experimental system, which uses a reporter line [murine T cell hybridoma transduced with rat/mouseCD28 (53/4) and human V γ 9V δ 2 TCR] to test activation by HMBPP, requires much higher concentrations than for PBMC assays involving primary V γ 9V δ 2 T cells as responders and various types of blood cells

(mainly monocytes) as stimulatory cells. Nevertheless, it is important to emphasize that there is no evidence that high concentrations of HMBPP have adverse effects on APC or responders.

The use of 293T cells and derivatives (BTN3A-ko cells and BTN3A transductants) as stimulatory cells allow us not only to test different BTN3A variants and combinations thereof, for their capacity to induce PAg-mediated activation but also to correlate this to features of the BTN3A molecules such as trafficking, intermolecular interaction, membrane mobility and surface expression, which would simply be impossible using “natural” stimulatory cells such as monocytes.

As a readout for V γ 9V δ 2 TCR -mediated activation we used IL-2 production by a reporter line. This line has the advantage over the use of primary V γ 9V δ 2 T cells is that it does not respond to multiple signals, such as (partially) TCR-independent activation, for example involving ligation of NK-cell receptors, various co-receptors and cytokines that stimulate primary V γ 9V δ 2 T cells. Furthermore, in contrast to human T cells (including T cell lines such as Jurkat), our reporter line does not express BTN2A1 and BTN3A molecules, preventing cross- and auto-presentation of HMBPP ¹. Although such effects have prompted some researchers to use aminobisphosphonate (ABP)-primed 293T cells as stimulators instead of using HMBPP as stimulus ²⁻⁵, a concern with such an approach is that ABP (unlike HMBPP) are known to induce proapoptotic events in target cells at high concentrations. Furthermore, our 53/4-based human or alpaca V γ 9V δ 2 TCR expressing reporter line(s) have been proven to mirror results on TCR- and BTN-dependency obtained with primary V γ 9V δ 2 T cells. As such they were instrumental in elucidating the mandatory role of BTN3A1 and BTN2A1 for V γ 9V δ 2 TCR-mediated activation by PAg, ABP and mAb 20.1 as well the contribution of non-PAg-binding BTNs (BTN3A2 and BTN3A3) to primary V γ 9V δ 2 T cell responses ^{2,6-11}. They also reproduced findings on an unexpected non-responsiveness of V γ 9V δ 2 TCR (D1C55) transgenic cells to mAb 20.1 ^{12,13}.

A general problem when using ABP-pulsed 293T cells as APC and V γ 9V δ 2 T cell lines or PBMC containing V γ 9V δ 2 T cells as responders to investigate parameters controlling V γ 9V δ 2 TCR-mediated activation lies in the mode of action of ABP that involves several parameters being altered compared to that involving HMBPP, in spite of the fact that the ultimate consequences are very similar (BTN-dependent activation of the V γ 9V δ 2 TCR). ABP acts indirectly by inhibiting FDPS activity with consequences for the entire cellular isoprenoid metabolism, potentially impacting connected events such as inhibition of isoprenylation of small GTPases and accumulation of IPP and of pro-apoptotic IPP derivatives. Depletion of FPP and downstream isoprenoids and sterols by ABPs is toxic, so there is only a narrow range of concentrations at which these compounds stimulate T cells without toxicity. Furthermore, ABPs requires the transport into the cells (probably by pino- or phagocytosis) as well within the cell by SLC37A3 ¹⁴. Despite this, we included experiments with ABP and 293T cells and primary V γ 9V δ 2 T cells as responders to reproduce published data on the contribution of non-PAg-binding BTN3As (BTN3A2 and BTN3A3) ¹⁵ and these experiments validate the general suitability of our experimental system. We also tested for V γ 9V δ 2 T cell response to ABP pulsed of 293T and 293T derivatives and for ABP induced BTN3A membrane immobility and confirmed and extended published data.

2. They should also comment on how HMBPP added exogenously to the cell culture can activate the intracellular B30.2 domain. How does HMBPP get into the cell? Is the entrance into 293T cells equally efficient as into “normal” cells (e.g. monocytes)?

So far it is unclear how HMBPP is transported into and within the cell, but it is clearly an energy dependent mode of transport which can be overcome with membrane penetrating HMBPP prodrugs^{16,17}. It is very likely that the efficiency of HMBPP transport into and potentially within the cell varies between cell types and thereby it might contribute to differential stimulatory capacity of cell types such as those outlined above. Nevertheless, since a consistent cell type (293T and 293T derivatives) was always used for stimulations, this is not relevant for interpretation of our study, which aims to clarify BTN3A features controlling V γ 9V δ 2 TCR-mediated activation. We therefore did not discuss this parameter of control of PAg-response to avoid distraction of the reader from the subject of our paper, which is the mode of function BTNs in V γ 9V δ 2 TCR-mediated activation.

Minor:

1. Line 146: should read “stronger” rather than “strong”?

This is now changed, at Line 150.

2. Line 579: Here they used BrHPP, while in all other assays they used HMBPP. Please explain abbreviation BrHPP and state the source.

We formerly used BrHPP to generate V γ 9V δ 2 T cell lines and mentioned this erroneously in the Mat. and Methods. We have now corrected this to HMBPP, in line 651.

3. Line 987: CRR should read CDR

This is now corrected in line 1086.

4. Lines 682-690 (references): I would suggest to include more recent papers, e.g. DOI: 10.1038/s41423-020-0504-x DOI: 10.1038/s41571-022-00722-1

These newer references have now been incorporated as suggested (Ref. 1 and 3).

Reviewer #2 (expert in immunogenetics of antigen presentation):

We thank the reviewer for his/her very positive comments and also for crediting our efforts to address published and unpublished as well as disputed work. We think that this is necessary to finally develop a unified model of PAg-induced $\gamma\delta$ T cell

activation to which we aim to contribute with this paper. Below you will find our answers to the specific comments and at the end also on the model shown in Fig. 6.

This manuscript explores aspects of the interaction between the juxta-membrane (JM) regions of butyrophilin (BTN) 3A1, 3A2 and 3A3 in homo- and heterodimers as assessed by cell surface expression, stimulation of an appropriate gamma-delta T cell reporter line and by selective mutation and structural modelling, along with the requirements for the location of V and B30.2 domains (Figs. 1-4). In addition, the authors examine the effect of a non-stimulatory phosphoantigen (PAg) on the interaction of the intracellular regions of 3A1 and 2A1 (Fig. 5). On this basis, the authors propose an overarching model to explain many disparate findings in the literature and in their own manuscript (Fig. 6).

To a reviewer familiar with the general outlines of BTN research over the last years, but without detailed knowledge of some recent findings (including the latest papers, preprints, manuscripts in preparation and personal communications cited in the text), the present manuscript seems to add important new experiments and interpretations to the growing literature about BTN-family function and mechanism, and contributes a model incorporating as much information as possible. However, there are some concerns to be addressed.

In a general sense, the results in Figs. 1-3 are detailed and convincing. The weakest part of the manuscript (in the opinion of this reviewer) is Fig. 4, in which the authors pick out one sequence feature from the JM region, then test sequence swaps (which are good experimental data) and finish with a series of structural predictions which are reasonable but only speculation without further experimental verification. Moreover, the authors do not explain why they chose this particular sequence feature and indeed point out that other parts of the JM regions must be important, without testing other sequence portions.

We thank the reviewer for the appreciation on the quality of the Figs. 1-3. As reviewer mentioned, Figs. 1 – 3 provided the important insight and perspective on the role of BTN3A-JM that is decisive on association of BTN3 molecules. At the same time, Fig. 4 and 5 are very much important as Fig. 4 presented evidence for the motif that plays a crucial role in JM homodimer and JM-heterodimer interactions as well as for the intrinsic contribution of the motif on the efficacy and function of the molecule. Likewise, it was established that PAg bound B30.2 domain of BTN3A1 interacts with BTN2A1 but Fig. 5 showed that nature of PAg is so crucial that it should be capable of binding to BTN3A1-B30.2 domain and should equally mediated the PAg-BTN3A1_B30.2 complex to interact with BTN2A1. We have addressed these comments in the later part.

Here the comments on Figs 4-6.

Fig. 5 is convincing work, but all but one panel is already reported in the paper of one of the co-authors (Hsaio et al 2022 Cell Chem Biol), with the new data in this manuscript examining the effect of 4-M-HMBPP, which is an important finding (T cell stimulation correlates not with PAg binding but with binding followed by B30.2 domain interaction between 3A1 and 2A1). (Parenthetically, the authors seemed to do a good job of crediting other labs for published and unpublished results leading up to the experiments in this manuscript.)

Finally, Fig. 6 is an excellent model, but there may be alternatives. For example, the model proposes that upon binding of PAg to the B30.2 domain of BTN3A1-3A2/3 heterodimer, their interaction of the V region of BTN2A1 with the TCR (which is not enough to stimulate Vgamma9Vdelta2 T cells) is supplemented by binding of BTN3A2/3 to the TCR which is enough for stimulation. The authors cite one short study from a company as well as unpublished data by others as evidence for T cell stimulation, but otherwise do not explain why they prefer this model to the idea that BTN2A1 binding to TCR is inhibitory for the T cell, and that PAg binding to BTN3A1 leads to BTN2A1 to stop binding the TCR, releasing the inhibition and allowing the TCR to bind an appropriate stress ligand on the target/antigen-presenting cell. Perhaps the answer is well-known to members of this scientific community, but it seems unclear at least to this reviewer.

We would like to thank the reviewer for his/her different yet interesting perspective, and we have discussed the possibilities of such alternative hypothesis at the later part.

The fact that the authors have written up and submitted their results is understandable, given the number of preprints, manuscripts in preparation and personal communications cited in the text, and of course they wish to get their story out before someone else says more-or-less the same things. However, is there a reason not to utilise the preprint server bioRxiv which would give their manuscript instant priority and (presumably) protect their submission to Nature Comms, while allowing them to finish the detailed experiments which would strengthen their conclusions? Of course, such assurances are between the communicating author and the editor, and are beyond the understanding of this reviewer.

To our understanding there is no principle difference between ResearchSquare and bioRxiv. Both are preprint servers referenced in data bases as PubMed but neither substitute for acceptance at a high quality journal such as Nature Communications. While many of the potential experimental avenues the reviewer highlights are entirely sensible avenues to explore, we feel that these could take a substantial time to complete and the current dataset is compelling enough to publish the core story imminently.

In addition, there are detailed concerns below.

Line 55, 222. “heteromers” and “homomers”, “homomers or heteromers”. These terms are used along with homodimers and heterodimers. Are these terms being used interchangeably, or is there a deeper meaning obvious to members of this scientific community (for example, homomers may include dimers, trimers and other higher order interactions). The simplest clearest usage is to be preferred, if possible.

We used the terms homodimer and heterodimer when discussing the data in context with the modeling efforts, which were done on dimers (Fig. 4) or when referring to work of others discussing dimers. Otherwise, we prefer the use of the terms homomer and heteromer since higher ordered complexes with unknown stoichiometry are not unlikely as indicated by the blue native gel data shown (Supplementary fig. 3b).

Lines 200-204, Figure 2b. “Under fixed-permeabilized conditions (right hand panels) FLAG-

Vdelta3A1 was detectable at the cell surface only if colocalizing with HA-3A1_A3JM (violet, right). In contrast to live conditions, clear colocalization of FLAG- Vdelta3A1 and HA-3A1 was observed in cytoplasmic vesicles. Notably, when FLAG-Vdelta3A1 was coexpressed with HA-3A1_A3JM, HA-tag was detected largely at the membrane, with hardly any detectable in cytoplasmic vesicles.” This reviewer saw lots of violet staining apparently at the cell surface in the center panel headed with FLAG-Vdelta3A1 plus HA-3A1, but struggled to see the very little violet staining in the far-right panel with FLAG-Vdelta3A1 plus HA-3A1_A3JM, which had lots of apparently intracellular vesicles with separate red and blue staining. These confocal images seem to be at odds with the flow cytometry in Fig. 2a, where only the FLAG-Vdelta3A1 plus HA-3A1_A3JM was well-expressed at the cell surface. Please explain. Were the panels switched, or is this reviewer simply misunderstanding the data?

In response to this point, this part of the results has been rewritten (lines 204-221) and Fig. 2 was slightly modified together with its complementary supplementary figure (Supplementary fig. 2a). In summary, the pictures have not been swapped but the permeabilization reduced/abolished the BTN3A1 surface staining and provoked misinterpretations. We believe that the new versions of the figures and text clarify this point substantially.

Lines 322-237, Figure 2f. “However, 3A1-CFP co-expressed with 3A1_A3JM-YFP revealed high FRET predominantly at the membrane (Fig. 2f, upper right) ... Even stronger FRET was observed at the membrane when FLAG-Vdelta3A1-CFP was co-expressed with 3A1_A3JM-YFP (Fig. 2f, lower right) ...”. In the two upper panels, the interactions are depicted as green (intermediate interaction according to the scale), but in the two lower panels, one is green and the other red (the highest level interaction). There is no indication of how representative these four panels (with two cells each) might be, and perhaps the authors might consider some form of quantitation from many such panels.

In response to this point, we have now included a quantitation of FRET data for whole cells (Supplementary fig. 2e), extending the more qualitative data of representative cells shown in the Fig. 2f and Supplementary fig. 2 (Lines 264-269).

Lines 244-245, Figure 2e. “... suggests the formation of heteromers in which their respective B30.2 domains are in FRET-able distance as predicted (Fig 2e).” The left-hand side of Fig. 2e with different models of V domain association is not discussed, and the right-hand side with different models of TM-JM-B30.2 association with regard to the ectodomains is not discussed, in particular to what has been learned from the FRET experiments. Did the authors wish to say that their findings of FRET-positive interactions are consistent with two models: the V-C interactions of ectodomains which lead to TM and JM far apart but B30.2 domains close, or with the C-C interaction of ectodomains which lead TM-JM-B30.2 all being close? Or what is intended here?

In response to this query, we have changed the text to clarify this point, in relation to lines 238-242, 253-256 and 264-269.

It is challenging to arrive at a firm conclusion that unequivocally correlates the structures of ectodomains and B30.2 domains purely based on FRET and other experimental data such as flow cytometry staining and IP. However, from the combined assessment of all the above experimental data, we have provided in the

Discussion our speculations on the possible ectodomain and cytoplasmic conformations. This is provided as a perspective to readers, and one that emphasises the need to develop improved experimental approaches to address the physiological structural associations of BTN3A ectodomains and B30.2 domains.

Lines 267-269, Figure 3d. "... we tested the effects ... on the cell surface immobility of 3A-molecules by FRAP (Fluorescence Recovery after Photobleaching)." Some introduction about what the meaning of immobility under these conditions would be very helpful for those who are not aware of what the point of the experiment might be.

We have now changed the text to clarify this point, in lines 294-299.

Line 281. "... two complexes of less than 44 kDa apparent MW." Molecular mass (Mr) is now considered the appropriate term (MW referring to weight which depends on gravity, not an intrinsic property of the molecule).

We have now changed MW to Mr.

Lines 292-296, Figure 3e. "To test whether similar effects were also observed for a homomeric ..., a mutant ... was generated co-expressed (Fig. 3e)." Despite looking and thinking, the point of showing Fig. 3e was unclear to this reviewer. Please explain a bit more in the text.

We have now included an explicit explanation of Fig. 3e in lines 327-333.

Lines 296-298, Figure 3g. "Stimulation was successfully detected with both the co-transductants where the PAg-binding site and wild type V-domain were located on different molecules (Fig. 3f-g) ...". Some aspects of this panel are not clear to this reviewer, and are not described in the text or figure legend. What is the difference between the dark blue and light blue V domains, the three colours of JM, and the arrow pointing up versus the "T" at the end of the brown line. Also, why is only one PAg shown bound to each of the two B30.2 dark orange domains; is this meant to imply that PAg can only bind to one of the two B30.2 domains? Finally, what is this panel supposed to represent, predictions or experimental results? If it is supposed to represent the true situation from experiments (implied from "observed outcomes" in the figure legend), where is the data?

An updated Fig. 3 is now provided with a new scheme at "g" incorporating labelling of BTN3A domains and PAg to WT B30.2 domains. The necessary information about the scheme is provided in figure legend. The panel represents the experimental results, and the respective data related to every scheme is provided in the figure legend.

Lines 305-308 and 313-314, Figure 4a. "We noted that the JM of BTN3A1 contains a positively charged lysine triplet (KKK) ... two negatively charged glutamic acid residues (ExE) at this position (Fig. 4a) ... suggesting other regions of the JM may also be involved in controlling cooperation and trafficking of associated 3A-molecules ...". There is no indication of why the authors focused on such a small feature of the sequences which differ in so many other places (and places in which the authors themselves describe as having effects).

Only one feature was examined functionally, and all the rest is predicted models, without any supporting experimental data, but with a huge amount of speculation. Moreover, there is no consideration of what conformational changes might occur when PAg binds to the B30.2 domain or when the 3A1 B30.2 domain binds the 2A1 B30.2 domain. A full set of experiments swapping sections or scanning by alanine/serine substitution would seem to be warranted, or at the least functional and structural predictions from swapping the other sections mentioned. Perhaps the authors can justify these experiments as the starting-point for a full investigation, if the rationale for picking this little sequence feature can be explained.

Firstly, we inspected BTN3 molecules specifically looking for differences between BTN3A1 on the one hand and BTN3A2, BTN3A3 and alpaca BTN3A on the other hand, based on previous observations that human BTN3A1 preferentially formed heterodimers with other BTN3A molecules, and that these other BTN3A molecules were lacking in alpaca, which encoded a single BTN3A molecule. This approach is principally different from previous studies which concentrated on BTN3A1 features and/or on a hypothesized coiled-coil structure of homodimers of BTN3A1, or other BTNs and BTNLs but did not address why BTN3A-heterodimers or alpaca BTN3A are superior in mediating PAg-sensing over BTN3A1 homodimers ^{3,5,18-21}.

The specific region tested in Fig. 4 drew our attention since even by visual inspection of the aa sequence it was extremely striking that BTN3A1 JM contains a stretch of five positively charged aa (RKKKR) which intuitively would be expected to affect dimerization in this region, particularly as these positions in other BTN3s contain two negatively charged amino acids (xExEx). Two additional factors justified our exploration of these amino acids.

Firstly, the JM region of BTN3 is thought to form a coiled coil structure, whereby inter-helical interactions are likely critical to the strength and specificity of intermolecular pairing. Based on the existing structural database of coiled coil domains, such interactions can be either of a hydrophobic or electrostatic nature. Secondly, there is ample precedent for interhelical electrostatic interactions dictating preference for homo or heterodimer pairing in native coiled coils, with oppositely charged pairings stabilizing, and similarly charged repulsive pairs destabilizing coiled coil conformation. In addition, engineering in electrostatic attractions over repulsions has been key to the *de novo* design of coiled-coils that favor heterodimer over homodimer formation ²²⁻²⁴. Therefore, the idea that differential charges in the RKKKR/xExEx coiled coil region of BTN3 molecules might preferentially favor heterodimerization is entirely plausible. Furthermore, while the reviewer is correct to point out the many other sequence differences between BTN3 molecules, these JM charged residues could be regarded as a 'smoking gun' in terms of providing a clue as to the region of BTN3A molecules responsible for driving heterodimerization

Consistent with these suspicions, as shown in Fig. 4, replacing these amino acids in the JM of BTN3A3 with those of 3A1 converts BTN3A3 to a BTN3A1-like molecule in terms of cell surface expression in complexes with BTN3A1VΔ and reduced its stimulatory capacity to that of BTN3A1. Therefore, this data supports our hypothesis that the JM region controls interaction between BTN3As and their functionality. The aim of the modeling was to provide a molecular rationale for our experimental data that emphasize the importance of this region in heterodimerization, and secondly, following our experimental result that swapping the RKKKR of BTN3A1 with the

corresponding BTN3A3 region is not sufficient to create a “superBTN3”, to explore candidate amino acids outside this charged motif that may also control JM mediated interaction, to be tested in future studies.

In alignment with the above points, we have changed the text and figure legend to make our intentions in this section clear and justify our focus on this charged region (lines 342-350).

Lines 284-285, Figure 5. “Here, we examined the ability of 4-M-HMBPP to support this interaction.” The text was totally opaque to this reviewer (including the description of “titrated BTN2A1 intracellular domain) until the figure was examined. Perhaps the experiment would be clearer if the sentence was modified to say “Here, we examined the ability of 4-M-HMBPP to support this interaction by expressing and purifying the full intracellular domains of BTN3A1 (construct BFI) and BTN2A1 (construct ID271), and then using isothermal calorimetry (ITC) to measure association.” Also, how was “Titration of 960 microM 4-M-HMBPP into the buffer” (panel a) carried out, what is meant by “Mole Ratio” on the x-axis of panel a, and how was the concentration of 4-M-HMBPP for the rest of the panels decided? Also, rather than simply a, b, c, d and e above the panels in the figures, the authors could allow the readers to understand much more easily if some descriptive title for each panel was shown (much like the other figures).

We thank the reviewer for his/her helpful suggestion, revised the text to clarify the meaning of this section (lines 423-429) and generated a new version of Fig. 5 that contains the descriptive titles as suggested.

Lines 399-400. “Firstly, it demonstrates a crucial role for the V-domain for cell surface expression of BTN3A molecules.” It seemed unclear to this reviewer whether the authors meant a particular 3A chain, or the heterodimers/heteromers. Perhaps the sentence might be “Firstly, it demonstrates a crucial role for the V-domain for cell surface expression of BTN3A chains.” or “Firstly, it demonstrates a crucial role for at least one V-domain for cell surface expression of BTN3A molecules.”

We have now corrected this line (444-445).

Lines 497-502, Figure 6. “The function of the BTN2A1-V domain would be to recruit the TCR by binding to the CDR2 and HV4 regions of the TCRgamma chain ... while other CDRs might additionally interact with the newly formed BTN2A1-BTN3a complex that is in line with the finding of the Willcox research group (Willcox, C. R. et al 2023, in preparation).” The authors do not mention an alternative model in which BTN binding to the TCR is inhibitory; do they believe that this model of inhibition is no longer viable? If the data in this manuscript contributes to the choice between models, should this be considered in the discussion.

We generated a new Fig. 6 which integrates findings of our recent paper on BTN3A1-BTN2A1-V domain interaction ²⁵ with the following key findings of this paper, i) the unique topology of extracellular domains of BTN3A-heteromers in a composite TCR-ligand containing BTN2A1, ii) the pivotal role of charged residues in

the BTN3A-JM for formation of coiled coil structures in BTN3A-heteromers and iii) finally the interaction between the PAg-loaded BTN3A1 B30.2 domain with the B30.2 domains of the BTN2A1 homodimer.

In response to the remarks on alternative models on suppression of the $\gamma\delta$ T cell response by BTN-TCR interaction, we are not aware of a publication suggesting such a model. Indeed, in our hands BTN2A1 expressed by rodent cells is stimulatory⁷ and it does not interfere with PAg-induced activation when expressed in RAJI cells (Herrmann and Karunakaran, unpublished data) although we have preliminary evidence that this might vary between experimental systems. We therefore included in the discussion a small paragraph that BTN2-BTN3 interaction occurring either constitutively or PAg-induced might release ($\gamma\delta$) T cells from BTN-induced inhibition (lines 557-583).

Reviewer #3 (expert in butyrophilins):

This article finetunes some aspects of the cooperation between different butyrophilin proteins. Overall for the community not so much news and in my perception more for a specialized journal and would benefit from some more stringent writing. Also the translational relevance of the findings is missing.

Resolving the mode of action of different BTN3 molecules is an important step in understanding PAg sensing, a process central to one of the most prevalent TCR reactivities in humans. In this regard, the different BTN3 molecules clearly have a major impact on the efficiency of PAg sensing, and the process can only be understood fully when their respective roles are defined in molecular terms. Also, more broadly, analogous $\gamma\delta$ T cell populations at distinct anatomical sites also critically recognize parallel BTN heteromers that also possess homologous intracellular B30.2 domains (e.g. human intestinal V γ 4 T cells, on BTNL3.8; mouse intestinal V γ 7 T cells on BTNL1.6), it is also highly plausible that our results are paradigmatic for how BTN-mediated activation of $\gamma\delta$ T cells may occur in general. Therefore, in alignment with the other reviewers view of our study (Reviewer 1 highlighting the fact it provides 'new and important information', Reviewer 2 'important new experiments and interpretations...and a model incorporating as much information as possible'), we firmly believe that our study advances the field significantly in an important area and justifies the study's publication in a journal of the profile of *Nature Communications*.

Some minor points picked up

- Fig. 1F: symbols not well visible

This has now been changed.

- Fig. 3D: Concentration of PAM and BT3 20.1 not mentioned (neither in figure, text or methods)

This is now rectified, with the concentration stipulated in the legend.

- Fig. 5: To me, this is the weakest part of the article.
 - o I would suggest to put headings to the graphs for a better overview. I think the line in (5a) is not correctly placed.
 - o Might be just me and the fact that I'm not used to interpreting this kind of experiments and graphs, but this part and figure took me quite long to really get through and understand why they can make certain statements. That BTN2A1 ID271 and BTN3A1 BFI are only interacting under pAg, has been shown in their reference already, and they didn't state any physiological relevance why to investigate 4-M-HMBPP instead of HMBPP. Is it an important competitor or very abundant?
 - o In their discussion they say "Notably, our ITC data are consistent with that model because we observe an n value near 1,.." (p. 18), but don't explain what the n value is and what conclusions you can draw from this.
 - o To me this part also comes a bit out of the blue and I don't see a clear link of this 4-M-HMBPP to the rest of the story.. Relevance of 4-M-HMBPP physiological conditions is also not being stated later on.

We have modified the introductory sentences of this part (lines 423-427), which together with a modified figure and table make our point clearer. In brief, 4-HMBPP is a synthetic HMBPP-derivative which stimulates very poorly but binds well to B30.2 domain of BTN3A1. The poor stimulatory capacity coincides with the impaired capacity of BTN3A1-B30.2 4-HMBPP complexes to bind to the BTN2A1 intracellular domain.

- Table 1 is missing a good table legend and in my opinion a more elaborate explanation.

This is now changed, with a legend included in Table 1.

- Please comment: Is there a reason why the experiments besides Stimulation assays have been performed in the absence of ABP?

The reasons why so few experiments were performed with ABP are explained in the last part of our response to comment 1 of reviewer 1.

** See Nature Portfolio's author and referees' website at www.nature.com/authors for information about policies, services and author benefits.

This email has been sent through the Springer Nature Tracking System NY-610A-NPG&MTS

Confidentiality Statement:

This e-mail is confidential and subject to copyright. Any unauthorised use or disclosure of its contents is prohibited. If you have received this email in error please notify our Manuscript Tracking System Helpdesk team at <http://platformsupport.nature.com>.

Details of the confidentiality and pre-publicity policy may be found here <http://www.nature.com/authors/policies/confidentiality.html>

Privacy Policy | Update Profile

DISCLAIMER: This e-mail is confidential and should not be used by anyone who is not the original intended recipient. If you have received this e-mail in error please inform the sender and delete it from your mailbox or any other storage mechanism. Springer Nature Limited does not accept liability for any statements made which are clearly the sender's own and not expressly made on behalf of Springer Nature Ltd or one of their agents. Please note that Springer Nature Limited and their agents and affiliates do not accept any responsibility for viruses or malware that may be contained in this e-mail or its attachments and it is your responsibility to scan the e-mail and attachments (if any).

Springer Nature Ltd. Registered office: The Campus, 4 Crinan Street, London, N1 9XW. Registered Number: 00785998 England.

References

1. Laplagne, C., Ligat, L., Foote, J., Lopez, F., Fournie, J.J., Laurent, C., Valitutti, S., and Poupot, M. (2021). Self-activation of Vgamma9Vdelta2 T cells by exogenous phosphoantigens involves TCR and butyrophilins. *Cell Mol Immunol* *18*, 1861-1870. 10.1038/s41423-021-00720-w.
2. Harly, C., Guillaume, Y., Nedellec, S., Peigne, C.M., Monkkonen, H., Monkkonen, J., Li, J., Kuball, J., Adams, E.J., Netzer, S., et al. (2012). Key implication of CD277/butyrophilin-3 (BTN3A) in cellular stress sensing by a major human gammadelta T-cell subset. *Blood* *120*, 2269-2279. 10.1182/blood-2012-05-430470.
3. Peigne, C.M., Leger, A., Gesnel, M.C., Konczak, F., Olive, D., Bonneville, M., Breathnach, R., and Scotet, E. (2017). The Juxtamembrane Domain of Butyrophilin BTN3A1 Controls Phosphoantigen-Mediated Activation of Human Vgamma9Vdelta2 T Cells. *J Immunol* *198*, 4228-4234. 10.4049/jimmunol.1601910.
4. Wang, H., and Morita, C.T. (2015). Sensor Function for Butyrophilin 3A1 in Prenyl Pyrophosphate Stimulation of Human Vgamma2Vdelta2 T Cells. *J Immunol* *195*, 4583-4594. 10.4049/jimmunol.1500314.
5. Wang, H., Nada, M.H., Tanaka, Y., Sakuraba, S., and Morita, C.T. (2019). Critical Roles for Coiled-Coil Dimers of Butyrophilin 3A1 in the Sensing of Prenyl Pyrophosphates by Human Vgamma2Vdelta2 T Cells. *J Immunol* *203*, 607-626. 10.4049/jimmunol.1801252.
6. Riano, F., Karunakaran, M.M., Starick, L., Li, J.Q., Scholz, C.J., Kunzmann, V., Olive, D., Amslinger, S., and Herrmann, T. (2014). V gamma 9V delta 2 TCR-activation by phosphorylated antigens requires butyrophilin 3 A1 (BTN3A1) and additional genes on human chromosome 6. *European Journal of Immunology* *44*, 2571-2576. 10.1002/eji.201444712.
7. Karunakaran, M.M., Willcox, C.R., Salim, M., Paletta, D., Fichtner, A.S., Noll, A., Starick, L., Nohren, A., Begley, C.R., Berwick, K.A., et al. (2020). Butyrophilin-2A1 Directly Binds Germline-Encoded Regions of the Vgamma9Vdelta2 TCR and Is Essential for Phosphoantigen Sensing. *Immunity* *52*, 487-498 e486. 10.1016/j.immuni.2020.02.014.
8. Cano, C.E., Pasero, C., De Gassart, A., Kerneur, C., Gabriac, M., Fullana, M., Granarolo, E., Hoet, R., Scotet, E., Rafia, C., et al. (2021). BTN2A1, an immune checkpoint targeting Vgamma9Vdelta2 T cell cytotoxicity against malignant cells. *Cell Rep* *36*, 109359. 10.1016/j.celrep.2021.109359.
9. De Gassart, A., Le, K.S., Brune, P., Agaoglu, S., Sims, J., Goubard, A., Castellano, R., Joalland, N., Scotet, E., Collette, Y., et al. (2021). Development of ICT01, a first-in-class, anti-BTN3A antibody for activating Vgamma9Vdelta2 T cell-mediated

- antitumor immune response. *Sci Transl Med* 13, eabj0835. 10.1126/scitranslmed.abj0835.
10. Rhodes, D.A., Chen, H.C., Williamson, J.C., Hill, A., Yuan, J., Smith, S., Rhodes, H., Trowsdale, J., Lehner, P.J., Herrmann, T., and Eberl, M. (2018). Regulation of Human gammadelta T Cells by BTN3A1 Protein Stability and ATP-Binding Cassette Transporters. *Front Immunol* 9, 662. 10.3389/fimmu.2018.00662.
 11. Fichtner, A.S., Karunakaran, M.M., Gu, S., Boughter, C.T., Borowska, M.T., Starick, L., Nohren, A., Gobel, T.W., Adams, E.J., and Herrmann, T. (2020). Alpaca (*Vicugna pacos*), the first nonprimate species with a phosphoantigen-reactive Vgamma9Vdelta2 T cell subset. *Proc Natl Acad Sci U S A* 117, 6697-6707. 10.1073/pnas.1909474117.
 12. Starick, L., Riano, F., Karunakaran, M.M., Kunzmann, V., Li, J., Kreiss, M., Amslinger, S., Scotet, E., Olive, D., De Libero, G., and Herrmann, T. (2017). Butyrophilin 3A (BTN3A, CD277)-specific antibody 20.1 differentially activates Vgamma9Vdelta2 TCR clonotypes and interferes with phosphoantigen activation. *Eur J Immunol* 47, 982-992. 10.1002/eji.201646818.
 13. Vavassori, S., Kumar, A., Wan, G.S., Ramanjaneyulu, G.S., Cavallari, M., El Daker, S., Beddoe, T., Theodossis, A., Williams, N.K., Gostick, E., et al. (2013). Butyrophilin 3A1 binds phosphorylated antigens and stimulates human gammadelta T cells. *Nat Immunol* 14, 908-916. 10.1038/ni.2665.
 14. Yu, Z., Surface, L.E., Park, C.Y., Horlbeck, M.A., Wyant, G.A., Abu-Remaileh, M., Peterson, T.R., Sabatini, D.M., Weissman, J.S., and O'Shea, E.K. (2018). Identification of a transporter complex responsible for the cytosolic entry of nitrogen-containing bisphosphonates. *Elife* 7. 10.7554/eLife.36620.
 15. Vantourout, P., Laing, A., Woodward, M.J., Zlatareva, I., Apolonia, L., Jones, A.W., Snijders, A.P., Malim, M.H., and Hayday, A.C. (2018). Heteromeric interactions regulate butyrophilin (BTN) and BTN-like molecules governing gammadelta T cell biology. *Proc Natl Acad Sci U S A* 115, 1039-1044. 10.1073/pnas.1701237115.
 16. Hsiao, C.H., Lin, X., Barney, R.J., Shippy, R.R., Li, J., Vinogradova, O., Wiemer, D.F., and Wiemer, A.J. (2014). Synthesis of a phosphoantigen prodrug that potently activates Vgamma9Vdelta2 T-lymphocytes. *Chem Biol* 21, 945-954. 10.1016/j.chembiol.2014.06.006.
 17. Kilcollins, A.M., Li, J., Hsiao, C.H., and Wiemer, A.J. (2016). HMBPP Analog Prodrugs Bypass Energy-Dependent Uptake To Promote Efficient BTN3A1-Mediated Malignant Cell Lysis by Vgamma9Vdelta2 T Lymphocyte Effectors. *J Immunol* 197, 419-428. 10.4049/jimmunol.1501833.
 18. Gu, S., Sachleben, J.R., Boughter, C.T., Nawrocka, W.I., Borowska, M.T., Tarrasch, J.T., Skiniotis, G., Roux, B., and Adams, E.J. (2017). Phosphoantigen-induced conformational change of butyrophilin 3A1 (BTN3A1) and its implication on Vgamma9Vdelta2 T cell activation. *Proc Natl Acad Sci U S A* 114, E7311-E7320. 10.1073/pnas.1707547114.
 19. Sandstrom, A., Peigne, C.M., Leger, A., Crooks, J.E., Konczak, F., Gesnel, M.C., Breathnach, R., Bonneville, M., Scotet, E., and Adams, E.J. (2014). The intracellular B30.2 domain of butyrophilin 3A1 binds phosphoantigens to mediate activation of human Vgamma9Vdelta2 T cells. *Immunity* 40, 490-500. 10.1016/j.immuni.2014.03.003.

20. Salim, M., Knowles, T.J., Baker, A.T., Davey, M.S., Jeeves, M., Sridhar, P., Wilkie, J., Willcox, C.R., Kadri, H., Taher, T.E., et al. (2017). BTN3A1 Discriminates gammadelta T Cell Phosphoantigens from Nonantigenic Small Molecules via a Conformational Sensor in Its B30.2 Domain. *ACS Chem Biol* *12*, 2631-2643. 10.1021/acscchembio.7b00694.
21. Nguyen, K., Li, J., Puthenveetil, R., Lin, X., Poe, M.M., Hsiao, C.C., Vinogradova, O., and Wiemer, A.J. (2017). The butyrophilin 3A1 intracellular domain undergoes a conformational change involving the juxtamembrane region. *FASEB J* *31*, 4697-4706. 10.1096/fj.201601370RR.
22. Graddis, T.J., Myszka, D.G., and Chaiken, I.M. (1993). Controlled formation of model homo- and heterodimer coiled coil polypeptides. *Biochemistry* *32*, 12664-12671. 10.1021/bi00210a015.
23. Kohn, W.D., Kay, C.M., and Hodges, R.S. (1995). Protein destabilization by electrostatic repulsions in the two-stranded alpha-helical coiled-coil/leucine zipper. *Protein Sci* *4*, 237-250. 10.1002/pro.5560040210.
24. Mason, J.M., and Arndt, K.M. (2004). Coiled coil domains: stability, specificity, and biological implications. *Chembiochem* *5*, 170-176. 10.1002/cbic.200300781.
25. Willcox, C.R., Salim, M., Begley, C.R., Karunakaran, M.M., Easton, E.J., von Klopotek, C., Berwick, K.A., Herrmann, T., Mohammed, F., Jeeves, M., and Willcox, B.E. (2023). Phosphoantigen sensing combines TCR-dependent recognition of the BTN3A IgV domain and germline interaction with BTN2A1. *Cell Rep* *42*, 112321. 10.1016/j.celrep.2023.112321.

REVIEWER COMMENTS

Reviewer #1 (expert in $\gamma\delta$ T cell activation):

The authors have adequately addressed my previous comments

Reviewer #1 commenting on authors' rebuttal to Reviewer #3 (no longer available for second review):

4-M-HMBPP is not a natural ligand, it is a synthetic compound; as such, it is physiologically not relevant. BUT: it is used in the experiments depicted in Fig. 5 to make the very important point that binding of pAg (HMBPP or 4-M-HMBPP) is not enough to induce the subsequent interaction between 3A1 and 2A1; while 4-M-HMBPP clearly binds to B30.2 of 3A1, this is not sufficient to induce interaction of the two BTNs (and in line, 4-M-HMBPP has only very weak stimulatory activity).

So, from my point view, this is a very nice set of experiments which proves that only pAg which bind to 3A1 AND induce subsequent interaction with 2A1 are potent gd activators.

Reviewer #2 (expert in immunogenetics of antigen presentation):

I thank the authors for including some of the suggestions from the last review to make the paper more accessible to the non-butyrophilin expert. I remain convinced that this is an important advance with an interesting model at the end to inspire future experimentation. Having said that, I was still struggling to work my way through the many experiments in all the panels, even having reviewed the paper once.

It is a very complex series of experiments, for which I found myself trying to keep everything straight: expression, interaction and signalling dependent on V domains, JM regions and 30.2 domains of 3A1 and 3A2 or 3A3, and then, more-or-less out-of-the-blue, 30.2 domain interactions between 3A1 and 2A1. At the end of it all, I find I am still not sure why heterodimers between 3A2 and 3A3 are not expressed and functional.

Nor am I clear on whether co-expression of swap mutants in the JM (3A1_ETE with 3A3_KKK) would be expected to function as well as 3A1 co-expressed with 3A3.

I also don't really understand why 3A1 and 3A1_A3JM can both go to the surface without supporting 3A2 or 3A3 molecules, but 3A1_A3JM signals so much better than 3A1 in the absence of other supporting molecules (I do understand that 3A1 doesn't go to the surface as well and isn't expected to produce such stable heterodimers, but the defect seems to be more profound than these differences).

All this goes to say that I haven't really understood what the experiments and models say, but it might just be me.

1. Lines 200-204. "FLAG-V Δ 3A1 was detected around the nuclear periplasmic space and this remained the same when coexpressed with HA-3A1 or HA-3A1_A3JM (Fig. 2a, Fixed). However, FLAG-V Δ 3A1 was detected at the surface only with HA-3A1_A3JM with which it largely colocalized (violet), and not with HA-3A1. Notably, FLAG-V Δ 3A1 colocalization with HA-3A1 was detected around nuclear periplasmic space and few vesicles (Fig. 2a and Supplementary Fig. 2a)." Is Fig. 2b intended (Fig. 2a is flow cytometry)? Also, I find myself confused since the text says that "FLAG-V Δ 3A1 was detected [only] around the nuclear periplasmic space" which fits with the lack of red staining for live "FLAG-Vdel3A1" (the left hand panel of the left hand three in figure 2b), but the red appears to be on the outside of the cell in fixed "FLAG-Vdel3A1" (the left hand panel of the right hand three in figure 2b). Or is that big red outline actually the perinuclear staining and the panel is sized incorrectly? Something seems wrong.

1. Lines 205-207. "Similar observations were made under fixed-permeabilized conditions

(Supplementary Fig. 2a, right hand panels). Despite cellular expression at high levels, little FLAG-VΔ3A1 was detectable at the cell surface, and only if coexpressed with HA-3A1_A3JM (violet, right).” Again, I find myself confused, since I see lots of violet, presumably at the cell surface, in the fixed-permeabilised FLAG-Vdel3A1 + HA-3A1 (middle panel of the three right hand panels in Supp Fig. 2a), but not in the fixed-permeabilised FLAG-Vdel3A1 + HA-3A1_A3JM (the far right hand panels of the three right hand panels). Again my reading of the text doesn’t fit what I see in the figure; are the authors sure that they have not mixed up the two columns?

2. Lines 231-233. “Based on existing structures, we predicted possible positioning of B30.2 domains in relation to head-tail and V-shaped ectodomain conformations and postulated them for FRET outcomes (Fig. 2e).” I found this panel quite confusing. The ectodomains and B30.2 domains are connected (they are one protein), so why have they been separated in the panel. Unless I have missed something, the two extodomain conformations are not mentioned again; how do they fit with the model being developed by the author.

3. Line 241. “The full-length 3A1-CFP/VΔ3A1-CFP coexpressed with 3A1-YFP displayed no FRET...”. What is meant by “3A1-CFP/VΔ3A1-CFP”? Should this be “Neither full-length 3A1-CFP nor full-length VΔ3A1-CFP coexpressed with 3A1-YFP displayed any FRET...”

4. Lines 244-249. “Bringing together the lack of surface detection of FLAG-VΔ3A1 when coexpressed with 3A1 (Fig. 2a) and their non-FRETtable B30.2 domains, let us to suspect whether these protein complexes may form head-tail ectodomains and distantly positioned pattern “A” of B30.2 domains (Fig. 2e). On the other hand, 3A1-CFP co-expressed with 3A1_A3JM-YFP revealed high FRET predominantly at the membrane (Fig. 2f, upper right), and with the increased intensity with the 530nm-filter (Supplementary Fig. 2d).” What does the lack of movement to the cell surface have to do with head-to-tail versus V-shaped ectodomains? It appears from Fig. 2e that it is possible to have a V-shaped ectodomain without “FRETtable” B30.1 domains (pattern C?). Do patterns A and B not require head to tail, and patterns C and D require V-shape? So sorry for being confused, but if I am, perhaps others would be as well.

5. Lines 255-258. “Collectively assessing detectable FLAG-VΔ3A1 when coexpressed with 3A1_A3JM and their FRETtable B30.2 domains let us to postulate that these BTN3A complexes may appear as V-shaped ectodomain and associated B30.2 domain as in pattern “C” (Fig. 2e).” Was pattern D intended? The B30.2 domains look closer together in pattern D than pattern C. Or am I misunderstanding?

6. Lines 258-260. “Furthermore, analysis for the total FRET (cytoplasmic and membrane) also revealed a significant FRET with 3KO cells transduced with FLAG-VΔ3A1-CFP or FLAG-3A1-CFP cotransduced with 3A1_A3JM-YFP but not with 3A1-YFP (Supplementary Fig. 2e).” I thank the authors for quantitating their data in response to my concern in the last review, but I notice that the red colour in one panel of Fig. 2e is an outlier, not even represented in the quantitation shown in Fig. 2e (not one point on the graph showing a ratio near 3).

7. Lines 262-264. “On the contrary, co-expression of FLAG VΔ3A1-CFP or 3A1-CFP with 3A1_A3JM suggests the formation of heteromers in which their respective B30.2 domains are in FRET-able distance as predicted (Fig. 2e).” What is meant by “as predicted”?

8. Lines 264-268. “We hypothesized that an equivalent type of association occurs for the intracellular domains of VΔ3A1 or 3A1 when coexpressed with 3A2 but could not address this using the same methodology due to the different lengths of the intracellular domains and consequently of the adjacent fluorophores, which would confound FRET efficiency.” I find the rationale unclear. A diagram is shown and the authors have an expectation that the B30.2 domains will be far apart based on that diagram, but the reality might be that the B30.2 domains are able to contact anyway (the cytoplasmic tail bends), so was this rationale tested?

9. Lines 289-290. “Constructs with a 3A1JM displayed no increased immobilization whereas those with a 3A3JM did (Fig. 3d).” The increase may be “statistically significant”, but the quantity is negligible. I don’t notice any mention of such immobilisation in the Discussion, so does it mean anything?

10. Lines 297-302. "Furthermore, native gel electrophoresis of solubilized membrane extracts revealed very large 3A1-mC complexes when prepared with detergent Brij 96 and Triton X100 (Supplementary Fig. 3b). In contrast, membranes solubilized with digitonin, which binds to cholesterol, massively reduced the size of 3A1mC molecular complexes. In the presence of 3A2 and 3A3 these complexes were dissociated into two complexes of less than 440 kDa apparent Molecular mass (Mr)." Again, like the immobilisation, these phenomena are not related to any aspect of the model (unless I missed something important); does this relate the periplakin story from Trowsdale's lab? In the supplementary figure legend, there is no description of the labels "C1" and "C2"; are these the complexes mentioned in the text? Also, there is no description of what these gels are, western blots or what and if so, with what antibody, etc.

11. Lines 306-307. "So far, we showed that functional impairment of 3A-heteromer formation coincides with reduced stimulatory capacity." It is not clear to me what is meant by "functional impairment"; is it "impairment of function" which in English might be considered to be something quite different. Where is the evidence that heteromer formation has been impaired? Cellular expression, cell surface and cellular location, and stimulation were tested, but was heteromer formation tested? The native gels did not address the presence of butyrophilin heteromers.

12. Lines 354-357. "These efforts first highlighted the potential of human BTN3A1, BTN3A2, BTN3A3, and also the single alpaca isoform VpBTN3, to each form biophysically plausible homodimers via intermolecular coiled coil interactions, stabilized in each case by numerous polar and non-polar interactions at the interhelical molecular interface." It seems unreasonable to describe a couple of interactions per panel as "numerous".

13. Lines 405-408. "This provides a molecular explanation for the observation that introduction of the 283-285 ETE sequence of BTN3A3 into 3A1 is insufficient to confer substantially increased expression and functionality (Supplementary Fig. 4a-b)." This is a "potential molecular explanation", particularly given that there is quantitation, not even unbiased observations. Where are the models for 3A1_ETE against 3A1? Where is the co-expression of swap mutants (3A1_ETE plus 3A3_KKK)?

14. Lines 417-421. "We confirmed a robust binding interaction between 4-M-HMBPP and the BTN3A1 full intracellular domain (3A1 BFI) (Fig. 5c) ... Next, we titrated BTN2A1 intracellular domain (2A1 ID271) into 3A1 BFI." Where is the information to describe "3A1 BFI" and "2A1 ID271" (that is, without having to refer to the previous publication)? Are these two domains known to act as monomers or dimers in solution and, if so, by what criteria? Please ensure that this is described, since it comes up in the discussion. (Also, what is the meaning of "mole ratio" for panel b?)

15. Lines 436-437. "Such compensation of loss of function BTN3A1-V constructs by residual levels of BTN3A2/BTN3A3 isoforms could explain ... ". As before, does this refer to BTN3A2 OR BTN3A3 or BTN3A2 AND BTN3A3 isoforms?

16. Lines 438-441. "It may also explain why a human V γ 9V δ 2 TCR transductant (TCR-MOP) that does not react to HMBPP-pulsed 3KO cells transduced with an alpaca BTN3(V-C)-human intracellular domain chimera but gains responsiveness when the same construct was transduced into BTN3A1KO cells, suggesting that chimera comprising heteromers involving V domains of endogenous BTN3A2 and/or BTN3A3 may engage with the human TCR or permit its ligation by an associated ligand." Please fix this sentence.

17. Lines 479-483. "Intriguingly, rescue of surface expression of V Δ BTN3A1 as an indicator for successful formation of BTN3A-complexes coincided very well with molecular modeling results on forces determining stabilization of coiled coil structures formed by JM α -helices, which are reduced for 3A1 JM homodimers relative to BTN3A3 JM or alpaca BTN3JM homodimers, and strongly favor heteromeric BTN3A1 JM interactions with BTN3A2 or BTN3A3JM." Unless I missed the calculation of molecular forces somewhere in the manuscript, this is an enormous overstatement. A completely qualitative observation of "predicted" (calculated) model structures for which the weakest of correlations are extracted, with almost no experimental tests. In the last review, I

suggested that the approach of modelling could be justified as the basis for exciting new experimental studies, but that has not appeared in the text.

18. Lines 525-527. "Notably, our ITC data are consistent with that model because we observe an n value near 1, which may be expected if a dimer of BTN2A1 is interacting with a monomeric PAg-ligand-bound form of BTN3A1-B30.2." Up to this moment, there has been no description of an "n value"; is that intended to be the same as mole ration (also misused in the fig. 5b).

19. Lines 550-553. "Our current results extend this model substantially by clarifying the critical role heteromeric partners (BTN3A2 and BTN3A3) of BTN3A1 play in the PAg sensing process, in facilitating interactions that are topologically optimized to surpass the requirements to initiate TCR signaling (Fig. 6)." Actually, figure 6 shows no interaction of BTN3A with TCR for tonic T cell signalling, so presumably "initiate TCR signalling" means "for activation".

20. Lines 560-561. "Our unpublished data suggests also on BTN2A1 as immunosuppressive molecule." If I have understood correctly, that in addition to "tonic signal, homeostasis, thymic selection" in the left hand panel of Fig. 6, 2A1 could be delivering a suppressive signal (so, add to figure)?

21. Fig. 3. Why are the JM different colours in iii and iv (the former identical chains in a homodimer, the latter nearly so)?

22. Lines 993-994. "a 293T and 3KO transductants of 3A1, 3A3, 3A3_R381H, or 3A_JM chimeric constructs were cultured and tested as in A (n-3)." What is meant by "A"?

23. Lines 108-110. "Schematic presentation of predicted intermolecular signaling within the BTN3A proteins correlated to the observed outcomes in terms of 53/4 human V γ 9V δ 2 TCR reporter activation strength with antigen-presenting cells (3KO) expressing V Δ 3A2 and 3A1 (I, represents Fig. 1h), V Δ 3A1 and 3A2

24. 1011 (II, represents Fig. 1f) including the 3A-constructs mentioned in f ...". It seems to me that the appropriate words are "inferred intermolecular signalling" rather than "predicted intermolecular signalling." Also, should "I" and "II" be "i" and "ii" to fit with the figure.

25. Lines 1013-1015. "Orange line represent signaling outcome, activating complex presented by n-terminus arrow mark styled line and non-activating complex presented by n-terminus T-shaped line." "N-terminus"

26. Fig. 6. The diagram looks like PAg is binding to a 3A2 B30.2 domain. Is this intended? If not, then please fix the diagram so that the 3A1 B30.2 domain is shown to the B30.2 domain without the PAg being in the interface. The B30.2 domains could bind due to a conformational change, or have I misunderstood?

Reviewer #3 (expert in butyrophilins):
Absent.

RESPONSE TO REVIEWERS' COMMENTS

We thank all reviewers for their quick, thorough, and careful comments and suggestions. We also like to thank the editor for providing an opportunity to revise the manuscript. Reviewers comments are in *italics*. The response in standard letters. The changes are also tracked in the manuscript.

Reviewer #1

The authors have adequately addressed my previous comments

Reviewer #1 commenting on authors' rebuttal to Reviewer #3 (no longer available for second review):

4-M-HMBPP is not a natural ligand, it is a synthetic compound; as such, it is physiologically not relevant. BUT: it is used in the experiments depicted in Fig. 5 to make the very important point that binding of pAg (HMBPP or 4-M-HMBPP) is not enough to induce the subsequent interaction between 3A1 and 2A1; while 4-M-HMBPP clearly binds to B30.2 of 3A1, this is not sufficient to induce interaction of the two BTNs (and in line, 4-M-HMBPP has only very weak stimulatory activity).

So, from my point view, this is a very nice set of experiments which proves that only pAg which bind to 3A1 AND induce subsequent interaction with 2A1 are potent gd activators.

We thank the reviewer for their positivity about the 4-M-HMBPP experiments and share their view that these make a very critical point regarding the importance of BTN3A1-BTN2A1 association for $\gamma\delta$ T cell activation.

Reviewer #2 (expert in immunogenetics of antigen presentation):

I thank the authors for including some of the suggestions from the last review to make the paper more accessible to the non-butyrophilin expert. I remain convinced that this is an important advance with an interesting model at the end to inspire future experimentation. Having said that, I was still struggling to work my way through the many experiments in all the panels, even having reviewed the paper once.

It is a very complex series of experiments, for which I found myself trying to keep everything straight: expression, interaction and signalling dependent on V domains, JM regions and 30.2 domains of 3A1 and 3A2 or 3A3, and then, more-or-less out-of-the-blue, 30.2 domain interactions between 3A1 and 2A1. At the end of it all, I find I am still not sure why heterodimers between 3A2 and 3A3 are not expressed and functional.

We accept that the study is an in-depth analysis of BTN3 structure-function aspects but have aimed to make the flow of experiments as logical as possible for the reader, progressing as the paper continues from the BTN3 N-terminus to the C-terminus. Therefore, we initially present the significance of each isoform, followed by the analysis of the V-domain, JM regions and finally assess pAg binding to the B30.2 domain, collectively revising our understanding of BTN3A complexes.

Regarding the significance of the different BTN3 molecules and the Referee's confusion about whether heterodimers between BTN3A2 and BTN3A3 might be competent for sensing, we emphasized in the Introduction the primary importance of BTN3A1 in pAg sensing, as well as the structural differences between the three BTN3A members, which

includes the absence (BTN3A2) or function-abrogating mutation (BTN3A3) of the critical 30.2 domain uniquely present in BTN3A1. In addition, and consistent with this perspective, in the Results section we show (in Fig. 1c) that cells expressing BTN3A2 and BTN3A3 are unable to sense P-Ag in the absence of BTN3A1. Conversely, we show BTN3A1 requires the presence of BTN3A2 and/or BTN3A3 for optimal response since their absence resulted in very poor T cell activation (Fig. 1a and 1b). Due to the clear conclusions from these results, we understandably did not test 3KO cells overexpressing recombinant BTN3A2+BTN3A3, since these will be unable to activate.

Nor am I clear on whether co-expression of swap mutants in the JM (3A1_ETE with 3A3_KKK) would be expected to function as well as 3A1 co-expressed with 3A3.

While this is an interesting suggestion, we did not carry out this experiment, firstly because it was not proposed in the first review of the manuscript and therefore was not prioritized, and secondly, because we would expect similar/identical results to the result obtained from co-expression of wild-type BTN3A1+BTN3A3 (Fig. 1c and 1d) in the existing manuscript, and therefore the result would be unlikely to add much to the current data set.

I also don't really understand why 3A1 and 3A1_A3JM can both go to the surface without supporting 3A2 or 3A3 molecules, but 3A1_A3JM signals so much better than 3A1 in the absence of other supporting molecules (I do understand that 3A1 doesn't go to the surface as well and isn't expected to produce such stable heterodimers, but the defect seems to be more profound than these differences).

As the reviewer pointed out, compared to BTN3A1, BTN3A1_A3JM works more efficiently. Our study presented the following evidence. i) BTN3A1_A3JM traffics better to surface than BTN3A1 [Fig. 2a by FACS]; ii) The cytoplasmic retention of BTN3A1_A3JM is much lower than BTN3A1 [Fig. 3c by confocal microscopy] and enhanced membrane immobilization of BTN3A1_A3JM [Fig. 2c by FRAP]; iii) BTN3A3JM permits the B30.2 domain interaction but not BTN3A1JM [Fig. 2e via FRET]; iv) finally, modelling provided insights into unstable BTN3A1JM homodimers and stable BTN3A3JM homodimers. In all the above scenarios, BTN3A1JM and BTN3A3JM containing BTN3A molecules exhibited appreciable differences in terms of molecular or biological properties that are well correlated to the functional outcome. Nevertheless, the exact mechanistic detail that eventually results in such differential outcomes cannot be defined with the current study.

All this goes to say that I haven't really understood what the experiments and models say, but it might just be me.

Although we believe that in general, we provided a clear description of the results and their interpretation we have now made a number of changes in order to clarify these points raised by the referee and which hopefully make the paper more digestible for a general audience. These include the decision to remove elements largely of "historical" interest such as those referring to models mainly based on crystallographic studies (e.g. form of dimers: head to tail vs V shaped) and whose biological significance is still unclear. We believe that this streamlining of the manuscript will help readers to better understand the key points of this study.

1. Lines 200-204. "FLAG-VΔ3A1 was detected around the nuclear periplasmic space and this remained the same when coexpressed with HA-3A1 or HA-3A1_A3JM (Fig. 2a, Fixed).

However, FLAG-VΔ3A1 was detected at the surface only with HA-3A1_A3JM with which it largely colocalized (violet), and not with HA-3A1. Notably, FLAG-VΔ3A1 colocalization with HA-3A1 was detected around nuclear periplasmic space and few vesicles (Fig. 2a and Supplementary Fig. 2a).” Is Fig. 2b intended (Fig. 2a is flow cytometry)? Also, I find myself confused since the text says that “FLAG-VΔ3A1 was detected [only] around the nuclear periplasmic space” which fits with the lack of red staining for live “FLAG-Vdel3A1” (the left hand panel of the left hand three in figure 2b), but the red appears to be on the outside of the cell in fixed “FLAG-Vdel3A1” (the left hand panel of the right hand three in figure 2b. Or is that big red outline actually the perinuclear staining and the panel is sized incorrectly? Something seems wrong.

We apologize for an error in figure numbering, which the reviewer has correctly highlighted. The figure numbering of Fig.2a was incorrect and is now corrected to Fig.2b. Also, to assuage the doubt regarding FLAG-VΔ3A1 localization at the surface in fixed cell staining, a new image of 3KO FLAG-VΔ3A1, stained with the membrane dye BODIPY-FL to distinguish membrane and perinuclear staining by anti-FLAG, has now been added, replacing the earlier image.

1. Lines 205-207. “Similar observations were made under fixed-permeabilized conditions (Supplementary Fig. 2a, right hand panels). Despite cellular expression at high levels, little FLAG-VΔ3A1 was detectable at the cell surface, and only if coexpressed with HA-3A1_A3JM (violet, right).” Again, I find myself confused, since I see lots of violet, presumably at the cell surface, in the fixed-permeabilised FLAG-Vdel3A1 + HA-3A1 (middle panel of the three right hand panels in Supp Fig. 2a), but not in the fixed-permeabilised FLAG-Vdel3A1 + HA-3A1_A3JM (the far right hand panels of the three right hand panels). Again my reading of the text doesn’t fit what I see in the figure; are the authors sure that they have not mixed up the two columns?

We would like to restate that the fixed-permeabilized staining presented in Supplementary fig. 2a is labelled properly. The middle panel shows what seems to be colocalization of FLAG (red) and HA (blue) observed in violet, presumably reflecting colocalization in cytoplasmic vesicles and nuclear periplasmic space, whereby the membranes were largely disrupted. However, in the far-right panel, cytoplasmic vesicles showed staining only for FLAG-VΔ3A1 (red) and HA-staining of HA-3A1_A3JM at the membrane (blue) and a few spots of colocalization (violet) at the membrane but not in the cytoplasmic vesicles. This is further validated by FLAG VΔ3A1-CFP (cyan) coexpressed with 3A1-YFP or 3A1_A3JM YFP (red) whereby surface expressed FLAG was detected as a few spots at the membrane despite high FLAG VΔ3A1-CFP expression (Supplementary fig. 2c). Fixed permeabilized staining of FLAG and HA-tagged constructs and live imaging of CFP and YFP fusion constructs shows the distinct trafficking pattern of VΔ3A1JM molecules featuring large cytoplasmic retention, and it could be detected at the surface only when coexpressed with 3A1_A3JM YFP. However, in the updated version, we would like to leave out fixed-permeabilized staining since it is not offering any new essential information and confocal images of CFP-YFP constructs are providing the same information without any confusion (lines 212 – 237).

2. Lines 231-233. “Based on existing structures, we predicted possible positioning of B30.2 domains in relation to head-tail and V-shaped ectodomain conformations and postulated them for FRET outcomes (Fig. 2e).” I found this panel quite confusing. The ectodomains and B30.2 domains are connected (they are one protein), so why have they been separated in

the panel. Unless I have missed something, the two ectodomain conformations are not mentioned again; how do they fit with the model being developed by the author.

We understand the concern raised by the reviewer. Bearing in mind the interests of a general audience not familiar with the history of models of BTN3A dimers and function, we have decided to remove this model from the figure and the accompanying text related to the models and their interpretation. This simplifies the manuscript considerably and focuses attention on the main novel model suggested by the collective dataset, set out in the Discussion, and summarized in the Graphical Abstract (lines 253 -257; 268-271; 280-285).

3. Line 241. "The full-length 3A1-CFP/VΔ3A1-CFP coexpressed with 3A1-YFP displayed no FRET...". What is meant by "3A1-CFP/VΔ3A1-CFP"? Should this be "Neither full-length 3A1-CFP nor full-length VΔ3A1-CFP coexpressed with 3A1-YFP displayed any FRET..."

This has now been corrected as suggested (line 265).

4. Lines 244-249. "Bringing together the lack of surface detection of FLAG-VΔ3A1 when coexpressed with 3A1 (Fig. 2a) and their non-FRETtable B30.2 domains, let us to suspect whether these protein complexes may form head-tail ectodomains and distantly positioned pattern "A" of B30.2 domains (Fig. 2e). On the other hand, 3A1-CFP co-expressed with 3A1_A3JM-YFP revealed high FRET predominantly at the membrane (Fig. 2f, upper right), and with the increased intensity with the 530nm-filter (Supplementary Fig. 2d)." What does the lack of movement to the cell surface have to do with head-to-tail versus V-shaped ectodomains? It appears from Fig. 2e that it is possible to have a V-shaped ectodomain without "FRETtable" B30.1 domains (pattern C?). Do patterns A and B not require head to tail, and patterns C and D require V-shape? So sorry for being confused, but if I am, perhaps others would be as well.

BTN3A1 due to its A1JM region may not form stable homodimers and its JM region may not be closely associated due to its distinct amino acid composition (including critical KKK residues) however IP revealed that VΔ3A1 coprecipitated with BTN3A1. Therefore, the BTN3A1 homodimer interaction can still be formed via the IgC-like domain and B30.2 domains. Crystal structures of soluble proteins have shown V-shaped dimers and head-to-tail dimers of BTN3A1 ectodomains. Likewise, the BTN3A1-B30.2 domains are also capable of forming dimers shown by crystals. Simplistically, we presumed that if the JM of a homodimer did not associate closely it may favor head-to-tail ectodomain dimers, yet it could allow B30.2 domains to interact (pattern A and B). Conversely, If JM regions are closely associated, it could favor a V-shaped dimer formation (pattern C and D). However, the lack of trafficking to the surface may have nothing to do with ectodomain conformation and another possibility is that an ectodomain configured in a head-to-tail conformation may result in poor detection of the FLAG tag. However, despite these arguments, to simplify the manuscript, we have excluded such interpretations regarding what are ultimately speculative models, whilst retaining the focus on the data generated (lines 253 -257; 268-271; 280-285).

5. Lines 255-258. "Collectively assessing detectable FLAG-VΔ3A1 when coexpressed with 3A1_A3JM and their FRETtable B30.2 domains let us to postulate that these BTN3A complexes may appear as V-shaped ectodomain and associated B30.2 domain as in pattern "C" (Fig. 2e)." Was pattern D intended? The B30.2 domains look closer together in pattern D than pattern C. Or am I misunderstanding?

We apologize for the error, it should have been pattern D. Again, to simplify and focus the manuscript, the discussion of results in relation to the assumed model was removed as stated above.

6. Lines 258-260. *“Furthermore, analysis for the total FRET (cytoplasmic and membrane) also revealed a significant FRET with 3KO cells transduced with FLAG-VΔ3A1-CFP or FLAG-3A1-CFP cotransduced with 3A1_A3JM-YFP but not with 3A1-YFP (Supplementary Fig. 2e).” I thank the authors for quantitating their data in response to my concern in the last review, but I notice that the red colour in one panel of Fig. 2e is an outlier, not even represented in the quantitation shown in Fig. 2e (not one point on the graph showing a ratio near 3).*

We thank the reviewer for his/her interest in these data. The ratiometric images (Fig. 2e) and FRET quantification (Supplementary Fig. 2e) are totally independent quantifications though the raw data for both remain the same. In the ratiometric FRET images, we could establish the fact that the proteins (FLAG-VΔ3A1-CFP and 3A1_A3JM-YFP) interact mostly at the membrane [shift in color from CFP to green or red considered as high FRET]. In the FRET quantification data (Supplementary Fig. 2e), we quantified the total FRET change (YFP/CFP) in the whole cell, which also includes intracellular (VΔ3A1-CFP or 3A1-CFP) proteins, that did not interact with 3A1_A3JM-YFP as it's not available in cytoplasmic vesicles. The incidences of FRET are possible only at the membrane where VΔ3A1-CFP or 3A1-CFP could interact with 3A1_A3JM-YFP as the latter predominantly traffics to the membrane. So, the FRET values are not very high in FLAG-VΔ3A1-CFP+3A1_A3JM-YFP compared to FLAG-3A1-CFP+3A1-YFP or FLAG-VΔ3A1-CFP+3A1-YFP. Therefore, it did not reflect the ratiometric FRET data.

7. Lines 262-264. *“On the contrary, co-expression of FLAG VΔ3A1-CFP or 3A1-CFP with 3A1_A3JM suggests the formation of heteromers in which their respective B30.2 domains are in FRET-able distance as predicted (Fig. 2e).” What is meant by “as predicted”?*

As expected might have been the appropriate term. We removed “as predicted” as it was in reference to the model Fig.2e as stated above, the text related to the model has been removed (lines 287-289)

8. Lines 264-268. *“We hypothesized that an equivalent type of association occurs for the intracellular domains of VΔ3A1 or 3A1 when coexpressed with 3A2 but could not address this using the same methodology due to the different lengths of the intracellular domains and consequently of the adjacent fluorophores, which would confound FRET efficiency.” I find the rationale unclear. A diagram is shown and the authors have an expectation that the B30.2 domains will be far apart based on that diagram, but the reality might be that the B30.2 domains are able to contact anyway (the cytoplasmic tail bends), so was this rationale tested?*

BTN3A2 does not have a B30.2 domain, resulting in a short cytoplasmic tailed compared to BTN3A1 and BTN3A3, which possess comparable cytoplasmic tail lengths. Hence, as we attempted to outline in the text, the FRET system used in this paper to investigate different distances/positioning of the fluorescence dyes in BTN3A1 and BTN3A1-A3JM protein of identical lengths may not be an appropriate system to study the FRET of BTN3A1 and BTN3A2. However, in response to this comment, we have removed the text on BTN3A1 and BTN3A2 FRET as it may mislead the audience (lines 289-294).

9. Lines 289-290. *“Constructs with a 3A1JM displayed no increased immobilization whereas those with a 3A3JM did (Fig. 3d).” The increase may be “statistically significant”, but the quantity is negligible. I don’t notice any mention of such immobilisation in the Discussion, so does it mean anything?*

The most relevant comparisons are between 3A1 vs 3A1_A3JM and 3A3-R381H vs 3A3-A1JM-R38A1H, where the molecules differ only at the JM region and yet displayed a roughly 30% difference in the degree of immobilization, which we do not regard as negligible. In addition, the previous reports that have demonstrated a positive correlation of immobilization to stimulation were cited in this part of the Results section (lines 311 – 316).

10. Lines 297-302. *“Furthermore, native gel electrophoresis of solubilized membrane extracts revealed very large 3A1-mC complexes when prepared with detergent Brij 96 and Triton X100 (Supplementary Fig. 3b). In contrast, membranes solubilized with digitonin, which binds to cholesterol, massively reduced the size of 3A1mC molecular complexes. In the presence of 3A2 and 3A3 these complexes were dissociated into two complexes of less than 440 kDa apparent Molecular mass (Mr).” Again, like the immobilisation, these phenomena are not related to any aspect of the model (unless I missed something important); does this relate the periplakin story from Trowsdale’s lab? In the supplementary figure legend, there is no description of the labels “C1” and “C2”; are these the complexes mentioned in the text? Also, there is no description of what these gels are, western blots or what and if so, with what antibody, etc.*

The change in the migration pattern of BTN3A complexes on SDS PAGE has not been demonstrated elsewhere to our knowledge, and we have no evidence they relate to periplakin association. Whilst they do not relate directly to the model we ultimately propose and the periplakin was shown to bind to BTN3A1 (Rhodes et al 2015), in this study, we did not investigate the association of periplakin with the BTN3A complexes detected on the western blot. We found the changes in the migration patterns of BTN3A complexes on SDS PAGE coincide with altered cellular trafficking as well as functional efficacy in terms of T-cell stimulation (lines 328 – 335). The cell lysate on the western blot revealed by mAb20.1 detected one high Mr complex in BTN3A1 expressing cells and two relatively lower Mr complexes (labelled as C1 and C2) in BTN3A1+BTN3A2+BTN3A3 expressing cells. The experimental procedures, details of the reagents and description of the western blot were updated in the legend of Supplementary Fig. 3b.

11. Lines 306-307. *“So far, we showed that functional impairment of 3A-heteromer formation coincides with reduced stimulatory capacity.” It is not clear to me what is meant by “functional impairment”; is it “impairment of function” which in English might be considered to be something quite different. Where is the evidence that heteromer formation has been impaired? Cellular expression, cell surface and cellular location, and stimulation were tested, but was heteromer formation tested? The native gels did not address the presence of butyrophilin heteromers.*

The text was rewritten with the term heteromers replaced with complexes or molecules (lines 339 – 345), reflecting the fact that heteromer formation was not formally tested, whereas the formation of complexes was.

12. Lines 354-357. *“These efforts first highlighted the potential of human BTN3A1, BTN3A2,*

BTN3A3, and also the single alpaca isoform VpBTN3, to each form biophysically plausible homodimers via intermolecular coiled coil interactions, stabilized in each case by numerous polar and non-polar interactions at the interhelical molecular interface.” It seems unreasonable to describe a couple of interactions per panel as “numerous”.

The modelling efforts highlight the BTN3A1, BTN3A2 and BTN3A3 homodimers were indeed stabilized by numerous interactions along the length of the helix and located outside the KKK motif, which for example for the BTN3A1 homodimer span residues Q274, E275, Q288, R291, E292 (mediated inter-helical polar interactions including salt bridges); and F281, W295, M298, V306 and L309 (mediating interhelical hydrophobic interactions). However, the figure panels were merely used to demonstrate what is likely the critical part of the interface that results in differential stability of homo- versus hetero-dimers. The text is therefore accurate and does not require alteration (lines 392-295).

13. Lines 405-408. “This provides a molecular explanation for the observation that introduction of the 283-285 ETE sequence of BTN3A3 into 3A1 is insufficient to confer substantially increased expression and functionality (Supplementary Fig. 4a-b).” This is a “potential molecular explanation”, particularly given that there is quantitation, not even unbiased observations. Where are the models for 3A1_ETE against 3A1? Where is the co-expression of swap mutants (3A1_ETE plus 3A3_KKK)?

The reviewer raises a reasonable point, namely that the molecular explanation provided is based on modelling studies. We have therefore altered the text in this section of the manuscript to include this important caveat. Whilst a model of 3A1_ETE against 3A1 (as suggested by the reviewer) would be expected to highlight similar clashes around the 277/278 region as observed for the BTN3A1 homodimer, based on the inherent uncertainty of such hypothetical models, rather than extend their use by also including models of such mutant homo/heteromer coiled coils, we think a better approach is to simply retain the current explanation but include these methodological caveats.

In summary, **modelling studies suggest that interhelical interactions** outside of the 283-285 region preferentially destabilize BTN3A1 homomers relative to both BTN3A2/3 homomers, and also relative to heteromers involving BTN3A1 and BTN3A2/A3. This provides a **potential** molecular explanation for the observation that introduction of the 283-285 ETE sequence of BTN3A3 into BTN3A1 is insufficient to confer substantially increased expression and functionality (Supplementary Fig. 4a-b) (lines 442 – 447).

14. Lines 417-421. “We confirmed a robust binding interaction between 4-M-HMBPP and the BTN3A1 full intracellular domain (3A1 BFI) (Fig. 5c) ... Next, we titrated BTN2A1 intracellular domain (2A1 ID271) into 3A1 BFI.”. Where is the information to describe “3A1 BFI” and “2A1 ID271” (that is, without having to refer to the previous publication)? Are these two domains known to act as monomers or dimers in solution and, if so, by what criteria? Please ensure that this is described, since it comes up in the discussion. (Also, what is the meaning of “mole ratio” for panel b?)

The description of BTN3A1 BFI and BTN2A1 ID271 was provided in Table 1 and is now also included in the Results of Fig. 5 (lines 458 – 461) and modified Fig. 5 with labels and legend (lines 1146 – 1148). No peer-reviewed study has reported the interaction between the B30.2 domains of these proteins, although our collaborator’s previous study reports PAg induced interaction between BTN3A1 BFI and BTN2A1 ID271. Nevertheless, there is a preprint available on this topic (B30.2 domain interaction) and we have cited this. In the Discussion,

the relevant information from this preprint was mentioned briefly. We have corrected “mole ratio” to “molar ratio” in the figure.

15. Lines 436-437. *“Such compensation of loss of function BTN3A1-V constructs by residual levels of BTN3A2/BTN3A3 isoforms could explain ... ”. As before, does this refer to BTN3A2 OR BTN3A3 or BTN3A2 AND BTN3A3 isoforms?*

This has been corrected to indicate BTN3A2 and BTN3A3 isoforms (line 478).

16. Lines 438-441. *“It may also explain why a human V γ 9V δ 2 TCR transductant (TCR-MOP) that does not react to HMBPP-pulsed 3KO cells transduced with an alpaca BTN3(V-C)-human intracellular domain chimera but gains responsiveness when the same construct was transduced into BTN3A1KO cells, suggesting that chimera comprising heteromers involving V domains of endogenous BTN3A2 and/or BTN3A3 may engage with the human TCR or permit its ligation by an associated ligand.” Please fix this sentence.*

As reviewer suggested the sentence has been edited and split into two for increased clarity:

“It may also explain why a human V γ 9V δ 2 TCR transductant (TCR-MOP) that does not react to HMBPP-treated 3KO cells transduced a chimera composed of alpaca BTN3 (V-C) and human 3A1 (transmembrane-JM-B30.2 domain) but gains responsiveness when the same construct was transduced into BTN3A1KO cells. This suggests that in the latter scenario, chimera comprising BTN3A complexes involving V domains of endogenous BTN3A2 and/or BTN3A3 may engage with the human TCR upon PAg sensing by B30.2 domain of the chimera or permit its ligation by an associated ligand.” (lines 481 - 488).

17. Lines 479-483. *“Intriguingly, rescue of surface expression of V Δ BTN3A1 as an indicator for successful formation of BTN3A-complexes coincided very well with molecular modeling results on forces determining stabilization of coiled coil structures formed by JM α -helices, which are reduced for 3A1 JM homodimers relative to BTN3A3 JM or alpaca BTN3JM homodimers, and strongly favor heteromeric BTN3A1 JM interactions with BTN3A2 or BTN3A3JM.” Unless I missed the calculation of molecular forces somewhere in the manuscript, this is an enormous overstatement. A completely qualitative observation of “predicted” (calculated) model structures for which the weakest of correlations are extracted, with almost no experimental tests. In the last review, I suggested that the approach of modelling could be justified as the basis for exciting new experimental studies, but that has not appeared in the text.*

We accept this was a miswording in the original submission and have rephrased the relevant text to omit any mention of force determination, and to make the language more moderate. While we accept these are qualitative observations from modelling approaches, the combination with expression comparison between homo/hetero-dimers, and mutations thereof, provides strong orthogonal insights into the molecular mechanisms involved. We appreciate the reviewer’s enthusiasm for additional mutational work and have highlighted in the revised Discussion the potential for such exciting future follow-up studies to address these points.

“Intriguingly, rescue of surface expression of V Δ BTN3A1 as an indicator for successful formation of BTN3A-complexes coincided very well with molecular modelling **of** coiled coil structures formed by JM α -helices, which **suggest reduced stability of** BTN3A1 JM

homodimers relative to BTN3A3 JM or alpaca BTN3JM homodimers, **and favour** heteromeric BTN3A1 JM interactions with BTN3A2 or BTN3A3JM.” (lines 524 – 528)

Discussion point included in the text as below (lines 536 - 538):

“Future mutational and/or structural work focussed on the coiled coil region could shed light on the nature of such critical interhelical interactions”.

18. Lines 525-527. “Notably, our ITC data are consistent with that model because we observe an n value near 1, which may be expected if a dimer of BTN2A1 is interacting with a monomeric PAg-ligand-bound form of BTN3A1-B30.2.” Up to this moment, there has been no description of an “ n value”; is that intended to be the same as mole ration (also misused in the fig. 5b).

The n value represents the stoichiometry of B30.2 domains and molecules interacting; this has been clarified in the Discussion (line 574).

19. Lines 550-553. “Our current results extend this model substantially by clarifying the critical role heteromeric partners (BTN3A2 and BTN3A3) of BTN3A1 play in the PAg sensing process, in facilitating interactions that are topologically optimized to surpass the requirements to initiate TCR signaling (Fig. 6).” Actually, figure 6 shows no interaction of BTN3A with TCR for tonic T cell signaling, so presumably “initiate TCR signaling” means “for activation”.

The presumed BTN3A recognition by TCR in the presence of PAg could be inferred as a stimulus to “initiate TCR signaling”, which is known to be critical for PAg-induced T cell activation. As we don’t have the information on the minute details involved in presumed BTN3A-TCR interactions and precise downstream signaling events but nevertheless observe the final T cell activation, we labelled it as “Activation”, which represents the ultimate consequence of this interaction. We have edited the text to provide better understanding to the readers (lines 596 – 606).

20. Lines 560-561. “Our unpublished data suggests also on BTN2A1 as immunosuppressive molecule.” If I have understood correctly, that in addition to “tonic signal, homeostasis, thymic selection” in the left hand panel of Fig. 6, 2A1 could be delivering a suppressive signal (so, add to figure)?

Our unpublished data also suggest BTN2A1 can act as an immunosuppressive molecule, but this was based on analysis of BTN2A1-overexpressing antigen-presenting cells, and it would require an extensive series of experiments to validate this observation in a more physiological setting. We have therefore not incorporated these early findings into the final model outlined in this study, which serves as a take-home message for the readers.

21. Fig. 3. Why are the JM different colors in iii and iv (the former identical chains in a homodimer, the latter nearly so)?

The Fig. 3g scheme involving BTN3A complexes in iii and iv are composed of BTN3A3JM (turquoise blue) and other BTN3A complexes involving heteromeric JM regions, are shown in different blue colors (3A1JM – sky blue; 3A2JM – Dark blue) as mentioned in the legend.

22. Lines 993-994. *“a 293T and 3KO transductants of 3A1, 3A3, 3A3_R381H, or 3A_JM chimeric constructs were cultured and tested as in A (n-3).” What is meant by “A”?*

We apologize for the error of omitting details of the functional assay. This has now been corrected and information on the functional assay is included (lines 1088 – 1090).

23. Lines 108-110. *“Schematic presentation of predicted intermolecular signaling within the BTN3A proteins correlated to the observed outcomes in terms of 53/4 human V γ 9V δ 2 TCR reporter activation strength with antigen-presenting cells (3KO) expressing V Δ 3A2 and 3A1 (I, represents Fig. 1h), V Δ 3A1 and 3A2*

24. 1011 (II, represents Fig. 1f) including the 3A-constructs mentioned in f ...”. *It seems to me that the appropriate words are “inferred intermolecular signalling” rather than “predicted intermolecular signalling.” Also, should “I” and “II” be “i” and “ii” to fit with the figure.*

We think the comments # 23 and # 24 are connected as its related to Fig. 3 . This has been altered to “inferred intermolecular signaling”, and upper-case Roman numerals have been changed to lowercase, as suggested (lines 1103- 1105).

25. Lines 1013-1015. *“Orange line represent signaling outcome, activating complex presented by n-terminus arrow mark styled line and non-activating complex presented by n-terminus T-shaped line.” “N-terminus”*

This has now been corrected to be “N-terminus”, as suggested (line 1110).

26. Fig. 6. *The diagram looks like PAg is binding to a 3A2 B30.2 domain. Is this intended? If not, then please fix the diagram so that the 3A1 B30.2 domain is shown to the B30.2 domain without the PAg being in the interface. The B30.2 domains could bind due to a conformational change, or have I misunderstood?*

We believe that the reviewer may have confused with BTN3A2 and BTN3A1 in the diagram, as 3A2 does not possess a B30.2 domain, and probably is related to the fact that there is likely ‘crossing over’ of the helical segments of BTN3A1 and BTN3A2. Secondly, it is clear from the pre-print publication on B30.2 domain association between BTN3A1 and BTN2A1 (which we referenced) that the PAg is bound primarily by the BTN3A1 B30.2 domain but when bound is also contacted directly by the BTN2A1 B30.2 domain. Hence the PAg is at the interface. To aid understanding of the figure we have modified the picture incorporating more pronounced colors for BTN3A1 and BTN3A2 to make it easier for readers to differentiate the PAg-bound BTN3A1 B30.2 domain from the BTN3A1 B30.2 domain lacking BTN3A2.

Reviewer 2, suggestions made the paper much clearer and more comprehensive. He/she has pointed out the minute details which may confuse the general audience thereby not enabling them to grasp the important messages. Appreciating his/her judgements, we added a few details and removed a couple of figures to make the paper much clearer. We would like to extend our thanks to the reviewer for investing his/her valuable time in assessing our work and like to convey that his/her comments were really helpful and critical in improving the quality of the paper.

Reviewer #3 (expert in butyrophilins):
Absent.

REVIEWERS' COMMENTS

The remaining comments from Reviewer 2 were assessed editorially, as per their request.